# Prediction-Powered Causal Inferences

**Riccardo Cadei**[1], **Ilker Demirel**[2,†], **Piersilvio De Bartolomeis**[3,†], **Lukas Lindorfer**[1],
**Sylvia Cremer**[1], **Cordelia Schmid**[4], **Francesco Locatello**[1]

[1]Institute of Science and Technology Austria (ISTA)
[2]Massachusetts Institute of Technology (MIT)
[3]Department of Computer Science, ETH Zurich
[4]INRIA, Ecole Normale Supérieure, CNRS, PSL Research University
[†]*Equal contribution.*

## Abstract

In many scientific experiments, the data annotating cost constraints the pace for testing novel hypotheses. Yet, modern machine learning pipelines offer a promising solution—provided their predictions yield correct conclusions. We focus on *Prediction-Powered Causal Inferences* (PPCI), i.e., estimating the treatment effect in an unlabeled target experiment, relying on training data with the same outcome annotated but potentially different treatment or effect modifiers. We first show that conditional calibration guarantees valid PPCI at population level. Then, we introduce a sufficient representation constraint transferring validity across experiments, which we propose to enforce in practice in *Deconfounded Empirical Risk Minimization*, our new model-agnostic training objective. We validate our method on synthetic and real-world scientific data, solving impossible problem instances for Empirical Risk Minimization even with standard invariance constraints. In particular, for the first time, we achieve valid causal inference on a scientific experiment with complex recording and no human annotations, fine-tuning a foundational model on our similar annotated experiment.

## 1  Introduction

Artificial Intelligence systems hold great promise for accelerating scientific discovery by providing flexible models capable of automating complex tasks. We already depend on deep learning predictions across various scientific domains, including biology [Jumper et al., 2021, Tunyasuvunakool et al., 2021], medicine [Elmarakeby et al., 2021, Mullowney et al., 2023], sustainability [Zhong et al., 2020, Rolnick et al., 2022], and social sciences [Jerzak et al., 2022, Daoud et al., 2023]. While these models offer transformative potential for scientific research, their black-box nature poses new challenges, especially when used to analyze new experimental data. They can perpetuate hidden biases, which are difficult to detect and quantify, potentially invalidating any analysis resting on their predictions [Cadei et al., 2024].

Recent works proposed leveraging predictive models for efficient and still valid statistical inferences on partially labeled data [Angelopoulos et al., 2023a,b], training the predictor on the annotated sample and then estimating the statistical quantities of interest, including the model's predictions on the unlabeled sample. In this paper, we focus on *causal inferences from* (outcome) *unlabeled experiments*. We similarly aim to retrieve the missing labels from their complex measurements, i.e., entangled and high-dimensional, such as images or videos, through a predictive model. However, we consider training such a model from similar yet different annotated experiments, e.g., different treatments or effect modifiers. We refer to this problem as *Prediction-Powered Causal Inference* (PPCI), differing from classic Prediction-Powered Inference for its motivation, i.e., causal, and setting, i.e., generalization.

39th Conference on Neural Information Processing Systems (NeurIPS 2025).

To tackle this problem, we first characterize when a predictor is *valid* for a given PPCI problem. We show that (statistical) *conditional calibration* with respect to the observed outcome parents preserves identifiability and still guarantees consistent estimation, e.g., via Augmented Inverse Propensity Weighted (AIPW) estimator [Robins et al., 1994], for the treatment effect in the unlabeled experiment. However, the validity of a predictor may not transfer even between similar experiments and assuming an infinite training sample by Empirical Risk Minimization, even with standard invariance constraints [Arjovsky et al., 2019, Krueger et al., 2021]. We then propose a sufficient condition to enforce in the representation backbone of the predictor, *causal lifting* the model to transfer causal validity on the target experiment (together with other standard, yet idealized, assumptions).

We further propose a novel, simple yet effective implementation of such a constraint via loss reweighting, which we name *Deconfounded Empirical Risk Minimization* (DERM). To test it, we considered ISTAnt experiment [Cadei et al., 2024] (unique real-world benchmark for treatment effect estimation with complex measurements), ignoring the outcome annotations, and trained the predictive model over a new annotated experiment of ours with the same annotation mechanism, but different recording platform (lower quality) and treatments. We further validate and confirm the results on a synthetic manipulation of MNIST dataset [LeCun, 1998] by controlling the data-generating process and causal effect.

More broadly, this paper emphasizes the "representation learning" aspect of "causal representation learning" [Schölkopf et al., 2021], which has traditionally focused on the theoretical identification of a representation without concerning any specific task. In Bengio et al. [2013], good representations are defined as ones "*that make it easier to extract useful information when building classifiers or other predictors*." In a similar spirit, we focus on representations that make extracting causal information possible with some downstream estimator. We hope that our viewpoint can also offer new benchmarking opportunities that are currently missing in the causal representation learning literature and have great potential, especially in the context of scientific discoveries.

Overall, our main contributions are:

  i. formulation and foundational theory for **Prediction-Powered Causal Inferences**, i.e., prediction-powered causal estimands identification and estimators properties,

  ii. a **new constraint** to transfer causal validity across similar experiments, **causal lifting** neural representations with out-of-distribution fine-tuning, alongside a practical implementation (**DERM**),

  iii. **first valid** causal inference on a scientific experiment (ISTAnt) with complex measurements and **no human annotations**, by fine-tuning a model on our similar annotated experiment.

## 2 Prediction-Powered Causal Inference

In this Section, we formulate the PPCI problem, introducing the terminology and general identification and estimation results characterizing a *causally* valid outcome model.

---

**Problem:** Prediction-Powered Causal Inference (PPCI)

Let $(T, W, Y, X) \sim \mathcal{P}$ be random variables, where $T \in \mathcal{T}$ denotes a treatment assignment, $W \in \mathcal{W}$ the observed covariates, $Y \in \mathcal{Y}$ the outcome of interest, and $X \in \mathcal{X}$ a complex measurement capturing $Y$ information. Let $g : \mathcal{X} \to \mathcal{Y}$ be a fixed predictor of $Y$ from $X$.

Given an i.i.d. sample $\{(T_i, W_i, \_, X_i)\}_{i=1}^n \overset{i.i.d.}{\sim} \mathcal{P}$, where the outcomes are not observed directly, **identify and estimate** the average potential outcome under a treatment intervention[a], i.e., $\tau_Y(t) := \mathbb{E}[Y \mid \mathrm{do}(T = t)]$, or related estimands, **from the prediction-powered sample** $\{(T_i, W_i, g(X_i))\}_{i=1}^n$.

---
[a]We focus on population-level causal estimands, but the problem generalizes to group-level quantities too by additionally conditioning on a covariates subset.

---

Several scientific problems fit under this framework, where a human domain expert, trained on similar data, acts as a predictor $g$, elaborating and annotating the high-dimensional measurements, aiming to collect evidence to answer the overarching scientific question. Replacing this step with a deep learning model requires guarantees, which generally differ from standalone prediction accuracy metrics. Systematic errors in a specific subgroup can invalidate any causal reasoning [Cadei et al.,

2024, Cadei and Internò, 2025]. However, the PPCI potential comes from the fact that good outcome models can be learned also from similar experiments and transferred zero-shot, e.g., changing setup and treatment of interest, as long as the (true) annotation mechanism stays invariant. Such a generalization goal is the main motivation behind the investigation of this problem, going beyond (statistical) Prediction-Powered Inference [Angelopoulos et al., 2023a,b] that is designed instead for efficiency in-distribution. As a practical, real-world motivating problem, we aim to achieve valid Average Treatment Effect estimation on ISTAnt without expensive human annotations, but by annotating instead with foundational models fine-tuned over our annotated similar experiment, with different treatments and recording platform. See the problem illustration in Figure 1.

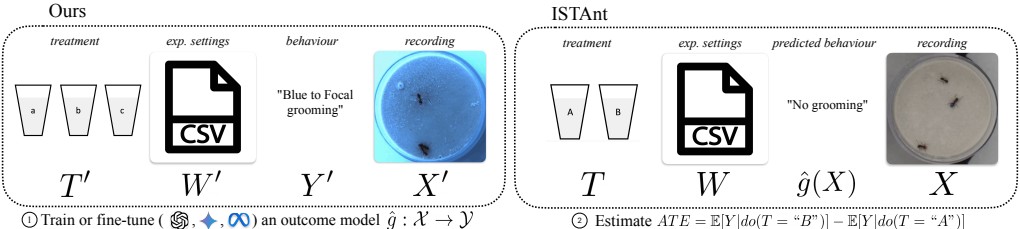

Figure 1: Pipeline for Average Treatment Effect inference on ISTAnt experiment without human annotations. We designed, recorded, and annotated our *similar* experiment with different treatments and recording platform and leveraged it to fine-tune a pre-trained model for imputation.

Before discussing the generalization challenges and our proposed mitigation, we formalize the properties a predictor requires to guarantee valid PPCI. Note that, in the following discussions, we always start by assuming identifiability of the average potential outcome(s), e.g., Randomized Controlled Trial. Indeed, if the problem has no solution even with access to the true outcomes, the corresponding PPCI problem loses meaning [1].

**Definition 2.1** (Valid Predictor). *An outcome predictor $g$ is valid for a PPCI problem if and only if the average potential outcome(s) of interest $\tau_Y(t)$ identify in a statistical estimand, i.e., a treatment function removing the do-operator, and $\tau_Y(t){=}\tau_{g(X)}(t)$.*

For example, given a PPCI problem $\mathscr{P}$ with binary treatment, let the average potential outcome upon intervening on the treatment value $t$ be identified via the adjustment formula by standard causal inference assumptions[2], i.e.,

$$\tau_Y(t) = \mathbb{E}_W[\mathbb{E}_Y[Y|T = t, W]] \tag{1}$$

where $W$ is a valid adjustment set. A predictor $g$ is valid for $\mathscr{P}$ if $\tau_Y(t){=}\tau_{g(X)}(t)$, i.e.,

$$\mathbb{E}_W[\mathbb{E}_Y[Y|T = t, W]] = \mathbb{E}_W[\mathbb{E}_X[g(X)|T = t, W]]. \tag{2}$$

Then, we want to characterize the testable statistical property a model has to satisfy to be valid.

**Definition 2.2** (Conditional Calibration). *Let $X, Y, Z \sim \mathcal{P}$ random variables with $X \in \mathcal{X}$ and $Y \in \mathcal{Y}$. A predictor $g : \mathcal{X} \to \mathcal{Y}$ is conditionally calibrated over $Z$ if and only if:*

$$\mathbb{E}[Y - g(X)|Z] \overset{a.s.}{=} 0 \tag{3}$$

**Lemma 2.1** (Prediction-Powered Identification). *Given a PPCI problem $\mathscr{P}$ with identifiable ATE (given the ground-truth annotations). If an outcome model $g : \mathcal{X} \to \mathcal{Y}$ is conditionally calibrated with respect to the treatment and the valid adjustment set considered for identification, then it is (causally) valid for $\mathscr{P}$.*

The lemma follows directly from backdoor adjustment, see proof in Appendix A, and offers a sufficient condition for validity. It is worth observing that being calibrated with respect to all the

---

[1]Unless an unobserved and valid adjustment set can be identified from the complex measurements $X$, but such a discussion is outside the scope of this paper.

[2]Consistency, exchangeability, and overlap.

observed covariates and not only the valid adjustment set is just a stronger sufficient condition, still holding the thesis. For example, in experiments where there are few discrete experimental conditions, it is safe to enforce/test it for all the covariates, without requiring a prior knowledge of which is the valid adjustment set.

> **Lemma 2.2** (Prediction-Powered Estimation). *Given a PPCI problem $\mathscr{P}$, with identifiable ATE (given the ground-truth annotations). If an outcome model $g : \mathcal{X} \to \mathcal{Y}$ is conditionally calibrated with respect to the treatment and a valid adjustment set, then the corresponding AIPW estimator over the prediction-powered sample preserves doubly-robustness and valid asymptotic confidence intervals, i.e.,*
>
> $$\sqrt{n}(\hat{\tau}_{g(X)} - \tau_Y) \to \mathcal{N}(0, V), \tag{4}$$
>
> *where $V$ the asymptotic variance.*

As before, the result is intuitive and now extends such a sufficient condition to preserve the property of an estimator. The proof, despite being technical, follows directly from classical AIPW results, and we present it in Appendix A. In practice, it is important to note that, while several estimators may exist for the same causal estimand, we need to rely on a valid predictor for the statistical estimand we identify the causal one with. For example, in a randomized controlled trial, we may identify the ATE with the associational difference, and therefore require calibration only with respect to the treatment; if, in practice, we want to use an estimator relying on covariates conditioning to improve efficiency, e.g., AIPW, we must require the outcome model to condition on all such covariates too. We still have not discussed how to enforce or even test such a property, without access to outcome annotations on the target experiment, and we start illustrating the fundamental generalization challenges in learning a valid outcome model.

## 2.1 Out-of-distribution generalization challenges

The main challenge in PPCI is training a valid outcome model to replace expensive and slow annotation procedures. It has to be expressive enough to process the complex input measurements, while being calibrated with respect to the experimental settings. Although modern pre-trained models are getting more and more capable and generalist [Bubeck et al., 2024], they still cannot guarantee calibration a priori [Chen et al., 2022]. As a human annotator learns an annotation procedure from similar experiments with the same measurement type (e.g., videos) and outcome of interest (e.g., ants' behaviours), we may aim to fine-tune pre-trained models on *similar* annotated experiments, leading towards scientifically valid expert models. For this purpose, let's first provide a notion of similarity among experiments.

**Definition 2.3** (Similarity). *Two PPCI problems $\mathscr{P}$ and $\mathscr{P}'$ are similar if their outcome-measurement mechanism is invariant, i.e.,*

$$\mathbb{P}(Y|X = x) = \mathbb{P}(Y'|X' = x) \quad \forall x \in supp(X) \cap supp(X') \tag{5}$$

This definition naturally generalizes to whatever measurement-outcome couple $(X, Y) \sim \mathcal{P}$ such that the annotation mechanism $\mathbb{P}(Y|X = x)$ is invariant. Note that our setting is different from the invariant causal predictions [Peters et al., 2016], as in our case $\mathbb{P}(Y|X = x)$ is not a causal mechanism that we want to identify using an invariance assumption over multiple environments, but rather $X$ is a measurement of $Y$ [Silva et al., 2006, Yao et al., 2025] (and thus a child of Y), potentially observed over a single environment $\mathscr{P}$. We can rely on this anticausal invariance whenever the same annotation protocol could be used for the two experiments (for example, humans trained to annotate $\mathscr{P}$ are capable of annotating $\mathscr{P}'$ without new training or instructions). Still, vanilla fine-tuning by standard Empirical Risk Minimization over *similar* experiments doesn't guarantee learning a valid outcome model due to peculiar generalization challenges raised by the spurious shortcuts appearing in anticausal prediction problems [Schölkopf et al., 2012, Peters et al., 2016], for example, from the visual appearance of certain effect modifiers. We distinguish here between statistical challenges with a finite training sample and potential issues even in the infinite sample regime.

**Finite training sample**   If the training sample is not representative of the full target population, the model may over-rely on its spurious conditional measurement-outcome correlation, e.g., some

irrelevant background information. Concretely, the measurements in such a subgroup are over-representing a certain treatment value or effect modifiers, e.g., up to violating positivity, which may bias any training and invalidate any causal downstream prediction-powered estimation.

---

**Example 1:** Background confounding

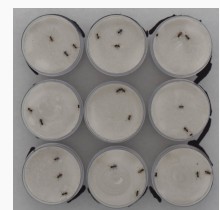 *Consider PPCI for ISTAnt, annotating only the first batch of nine videos for training. If a certain behaviour is underrepresented in a specific plate, e.g., because the plate is in the control group, the model can leverage such spurious shortcut in subsequent batches, relying on background information like the pen markings on the floor to identify the plate. This invalidates any downstream prediction-powered analysis, as future plates may be treated, but the model may assume they are not, matching the target statistics with the control group.*

---

**Infinite training sample**    Additionally, treatment information captured in the measurement that is not mediated by the outcome may be misleading for generalization, even in the infinite training sample setting. In other words, a double blind experiment needs to be blind also for the model, which should not be able to guess the treatment and other effect modifiers if not through the effect. Otherwise, the causal relationships in the training data can be wrongly transferred to the target experiment, potentially on different variables and distributions. For example, if an outcome value never happens in the training treatment, i.e., $supp(Y'|T' = t') \subset supp(Y')$, the model will wrongly transfer this information to a target population with a different treatment but with the same appearance, invalidating any causal downstream prediction-powered estimation there.

---

**Example 2:** Different treatments with similar appearance

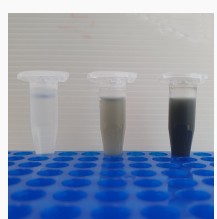 *Consider two experiments where the treatments are visually identical, discernible from the control, and differing in the effect, e.g., different pathogen suspension exposure with liquid residual in the measurements. While learning, an outcome model may over-rely on the training treatment effect, e.g., excluding the possibility of a specific behaviour, and wrongly transfer during PPCI on the target experiment. Note that this is not the case for our experiment and ISTAnt where the treatment is not visible, as for well-designed double blind experiments.*

---

## 3    Causal Lifting of Neural Representations

In the previous Section, we introduced and discussed PPCI and its peculiar challenges to learn a valid outcome model from training data out-of-distribution. In this Section, we introduce a new representation learning constraint to prevent such generalization challenges, and a corresponding implementation, tailored for our application of interest.

**Definition 3.1** (Causal Lifting constraint). *Given a PPCI problem $\mathscr{P}$, with identifiable ATE (given the ground-truth annotations), and a predictor $g = h \circ \phi : \mathcal{X} \to \mathcal{Y}$. The encoder $\phi : \mathcal{X} \to \mathbb{R}^d$ satisfies the Causal Lifting constraint if:*

$$I(\phi(X), Z|Y) = 0, \tag{6}$$

*where $Z = [T, \tilde{W}]$, and $\tilde{W} \subseteq W$ is a valid adjustment set for the ATE identification.*

The Causal Lifting constraint is a condition on the model encoder to prevent the possibility of leveraging unnecessary spurious outcome information from the visible experimental settings.

---

**Theorem 3.1** (Causal Lifting transfers causal validity). *Given two similar PPCI problems $\mathscr{P}$ and $\mathscr{P}'$, with identifiable ATE. Let $g = h \circ \phi : \mathcal{X} \to \mathcal{Y}$ be an outcome model for $\mathscr{P}'$ with $h$ Bayes-optimal, and assume that the representation transfers, i.e., $\mathbb{P}(\phi(X)|Y = y) = \mathbb{P}(\phi(X')|Y' = y)$. Then the Causal Lifting constraint on $\mathscr{P}$ implies $g$ is valid on $\mathscr{P}$.*

---

Theorem 3.1 offers a sufficient condition to transfer model validity from a training experiment to a target experiment by assuming the Causal Lifting constraint holds on target; see proof in Appendix A. It combines the strengths of modern deep learning models, able to extract sufficient representations on training tasks, i.e., $I(Y|\phi(X)) = I(Y|X)$, together with efficient fitting on limited data, i.e., Bayes Optimal classifier assumption. We remark that the assumption that the representation transfers is reasonable, since invariant annotation protocol commonly exists. In practice, the Causal Lifting constraint is required to hold on the target experiment, where we cannot get any guarantees without labels. However, referring to the generalization challenges examples in Section 2:

- in Example 1, the causal lifting constraint prevents the outcome model to leverage the spurious shortcuts from the background information by enforcing it to be position independent, i.e., background-agnostic,

- in Example 2, it prevents the outcome model to leverage the spurious training treatment effect information, by enforcing it to be treatment agnostic.

Despite the unverifiability of this assumption, we practically rely on the bet that transferring such property is an easier generalization task than transferring conditional calibration directly. We then propose to enforce the causal lifting constraint during training/fine-tuning *on the treatment and the valid adjustment set for the target experiment*, when also observed. If not observed but constant in the training experiment, the condition is satisfied by default (no signal for the model to learn such experimental settings). As pre-trained models over huge generic corpora get capable of detecting all visible information that can correlate with a prediction target, the causal-lifting constraint has to be enforced explicitly during transfer learning, forcing the artificial annotation process to be "blind" with respect to certain information. Similar to the Conditional Calibration assumption, if the outcome is well predictable from the measurement, enforcing the causal lifting constraint on more experimental settings than necessary does not hurt.

## 3.1 Deconfounded Empirical Risk Minimization

The Causal Lifting constraint is a conditional independence condition, and as such, enforcing it during training is a well-known problem in the Representation Learning literature. Different direct and indirect approaches have already been proposed, and in Section 4 we overview the major paradigms. Here, we propose a simple implementation that is tailored to common assumptions in scientific experiments (and that are true in ISTAnt), i.e., that experimental settings are low-dimensional and discrete [Pearl et al., 2000, Rosenbaum et al., 2010]. Other more general implementations are certainly possible if such assumptions do not hold, but this is beyond the scope of this paper. We leverage a resampling approach [Kirichenko et al., 2022, Li and Vasconcelos, 2019], reweighting the training distribution to simulate a distribution that has no correlation between outcome and experimental settings, while keeping the annotation mechanism invariant. We define the manipulated distribution of $Z^\star, Y^\star$ as function of $Z, Y$ and their joint distribution:

$$\mathbb{P}(Y^\star = y, Z^\star = z) := \begin{cases} 0 & \text{if } |supp(Y|Z=z)| = 0 \\ \overbrace{\frac{1}{|supp(Y|Z=z)|}}^{\mathbb{P}(Y^\star|Z^\star=z)} \cdot \overbrace{\frac{\text{Var}(Y|Z=z)}{\displaystyle\sum_{z'\in\mathcal{Z}} \text{Var}(Y|Z=z')}}^{\mathbb{P}(Z^\star=z)} & \text{otherwise} \end{cases} \tag{7}$$

for all $y \in \mathcal{Y}, z \in \mathcal{Z}$, where $supp(Y|Z=z) = \{y \in \mathcal{Y} : \mathbb{P}(Y=y|Z=z) > 0\}$. By design such joint distribution depends only on the experimental settings values, i.e., the conditional outcome distribution $\mathbb{P}(Y^\star|Z^\star = z)$ is uniform, while the experimental settings marginal $\mathbb{P}(Z^\star = z)$ weights more the observations with the least outcome-informative experimental settings (high conditional variance) over the training population, and ignores the most informative ones (low or null variance).

If the conditional outcome support is full for each not fully informative covariates value, i.e.,

$$supp(Y|Z=z) = supp(Y) \quad \forall z \in \mathcal{Z} : \text{Var}(Y|Z=z) > 0, \tag{8}$$

then, the conditional outcome distribution is constant and $Y^\star \perp\!\!\!\perp Z^\star$. It follows that an outcome model trained on such distribution has no signal from the outcome to learn the uncorrelated experimental settings, naturally enforcing the Causal Lifting constraint.

We then propose to train/fine-tune the factual outcome model minimizing the empirical risk, e.g., via Stochastic Gradient Descent, on such 'disentangled' distribution reweighting each observation $i$ by a factor:

$$w_i = \frac{\hat{\mathbb{P}}(Y^\star = y, Z^\star = z)}{\hat{\mathbb{P}}(Y = y, Z = z)}, \tag{9}$$

computed una-tantum before starting the training, estimating the joint training distribution by frequency (denominator), and the conditional variances by sample variance (nominator). We refer to this procedure as *Deconfounded Empirical Risk Minimization* (DERM).

Finally, we remark that when the condition in Equation 8 doesn't hold, it is not possible to specify a 'disentangled' joint distribution without ignoring the experimental settings, i.e., $\mathbb{P}(Z^\star = z) = 0$, where the conditional outcome's support is strictly contained in the marginal outcome's support on the training distribution. Our manipulated distribution will still consider these samples, but reducing their weight with respect to the predictivity of the observed experimental setting (approximately $\propto \mathrm{Var}(Y|Z = z)$). Therefore, some spurious correlations may still be learned, but this is now in a trade-off with ignoring a potentially substantial part of the reference sample. While this does not apply to our application of interest, tailored modifications of our joint distribution can be proposed case-by-case.

## 4 Related Works

**Prediction-Powered Inference**  The factual outcome estimation problem for causal inferences from complex observations was first introduced by Cadei et al. [2024]. We provide here its first formulation and corresponding foundational discussions and a valid approach to the problem, i.e., causal lifting pretrained neural representation. Particularly, we formalize what they describe as "encoder bias", and our DERM is the first proposal to their call for "*new methodologies to mitigate this bias during adaptation*" under similar yet different fine-tuning distribution. Demirel et al. [2024], Wang et al. [2024] already attempted to discuss similar generalization challenges with artificial predictions for causal inferences. However, they both ignored the main problem motivation to retrieve the latent outcomes from complex measurements, and not just weak signal from limited experiment settings, relying instead on unrealistic distributional assumptions. They ignored any entangled and high-dimensional measurement of the outcome of interest and assumed that a few experimental settings alone are sufficient for factual outcome estimation, together with support overlapping, i.e., without generalization, making any model application-specific and ignoring any connection with general-purpose foundational models. Let's further observe that their framework is a special case of ours when $X$ is interpretable and low-dimensional, and the target experiment is in-distribution. In contrast to the classic Prediction-Powered Inference (PPI) [Angelopoulos et al., 2023a,b], which improves estimation efficiency by imputing unlabeled in-distribution data via a predictive model, and recent causal inference extensions [Wang et al., 2024, De Bartolomeis et al., 2025, Poulet et al., 2025] relying on counterfactual predictions, PPCI focuses on imputing missing *factual* outcomes to generalize across similar experiments, yet out-of-distribution.

**Invariant representations**  Learning representations invariant to certain attributes is a widely studied problem in machine learning [Moyer et al., 2018, Arjovsky et al., 2019, Gulrajani and Lopez-Paz, 2021, Krueger et al., 2021, Yao et al., 2024a]. Practically, we aim to learn representations of $X$ that can predict the outcome $Y$, but are invariant to the experimental variables $Z$. In agreement with Yao et al. [2024a], we combine a sufficiency objective with our causal lifting constraint in the representation space, enforcing the suitable representation invariance. Several alternative approaches can be considered to enforce such conditional independence constraints: (i) Conditional Mutual Information Minimization [Song et al., 2019, Cheng et al., 2020, Gupta et al., 2021], (ii) Adversarial Independence Regularization such as Louizos et al. [2015] which modifies variational autoencoder (VAE) architecture in Kingma [2014] to learn *fair* representations that are invariant sensitive attributes, by training against an adversary that tries to predict those variables, (iii) Conditional Contrastive Learning such as Ma et al. [2021] whereby one learns representations invariant to certain attributes by optimizing a conditional contrastive loss, (iv) Variational Information Bottleneck methods where one learns useful and sufficient representations invariant to a specific *domain* [Alemi et al., 2016, Li et al., 2022]. We chose our DERM procedure as it matches well our application of interest, but other applications may prefer different conditional independence implementations.

**Causal Representation Learning**   In the broader context of causal representation learning methods [Schölkopf et al., 2021], our proposal largely focuses on representation learning applications to causal inference: learning representations of data that make it possible to estimate causal estimands. We find it in contrast with most recent works in causal representation learning, which uniquely focused on complete identifiability of all the variables or blocks, see Varici et al. [2024], von Kügelgen [2024] for recent overviews targeting general settings. The main exceptions are Yao et al. [2024a,b]. The former leverages domain generalization regularizers to debias treatment effect estimation in ISTAnt from selection bias. However, their proposal is not sufficient to prevent confounding when no data from the target experiment is given. The latter uses multi-view causal representation learning models to model confounding for adjustment in an observational climate application. In our paper, we also discuss conditions for identification, but we focus on a specific causal estimand, as opposed to block-identifiability of causal variables. As opposed to virtually all previous work in causal representation learning, our perspective offers clear evaluation and benchmarking potential on *real-world* scientific experiments – even in theoretically under-specified settings via the accuracy of the causal estimate, as also recently proposed by Yao et al. [2025].

## 5   Experiments

In pursuit of scientific validity, we evaluate our method (DERM) on the ISTAnt experiment [Cadei et al., 2024], and additional synthetic experiments controlling for the true causal effect.

**ISTAnt**   We performed a *similar* experiment to the ISTAnt experiment on our recording platform, considering three different treatments and collecting 44 annotated 30-minute videos, and we leveraged such annotated videos to fine-tune a pretrained model generalizing the predictions to ISTAnt and then estimating the ATE by AIPW on such prediction-powered sample. We considered the AIPW estimation on the (true) ISTAnt human annotated outcomes as the ground truth ATE (approximately $+40$ short clips with 2 clips per second, i.e., $\approx 20s$, in grooming time per video). Further details on the modeling choices, hyper-parameter, and fine-tuning are discussed in Appendix B. Figure 3 shows the differences in filming setup among the two experiments.

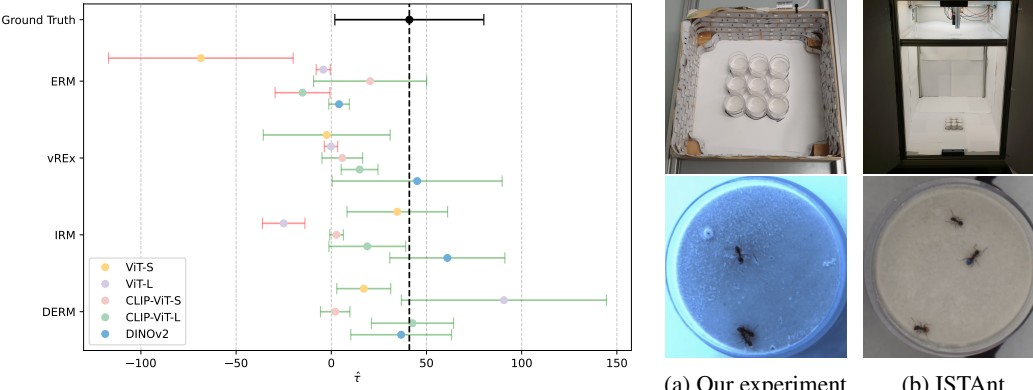

(a) Our experiment        (b) ISTAnt

Figure 2: PP-ATE inference on ISTAnt, fine-tuning different pre-trained encoders on our similar experiment, varying the training objective. Reported the 95% confidence intervals (via AIPW asymptotic normality) and compared against the same inference using the annotated outcomes. In green the correct confidence intervals (overlapping 33+%) in red the incorrect.

Figure 3: Filming box and example frame from our experiment and ISTAnt. The two experiments mainly differ in lighting conditions, treatments, experimental nests (wall height) and color marking.

Figure 2 reports the ATE estimation 95% confidence intervals per method-encoder (five different pre-trained backbones) by AIPW asymptotic normality, selecting per group the model across minimizing the treatment effect bias on reference. We trained multiple nonlinear heads, i.e., multi-layer perceptron, on top via (i) vanilla ERM, (ii) Variance Risk Extrapolation (vREx) Krueger et al. [2021], (iii) Invariance Risk Minimization Arjovsky et al. [2019] (IRM) and (iv) DERM (ours). As expected, fine-tuning via vanilla ERM leads to consistently underestimating/ignoring the effect (no validity guarantees). IRM and vREx, as proposed by Yao et al. [2024a], performs better, despite still being

occasionally invalid. DERM is the only method consistently providing valid ATE inference for all the pre-trained models, without ever estimating the effect with the wrong sign, unlike ERM, vREx and IRM.

**CausalMNIST**  We replicate the analysis on a colored manipulation of the MNIST dataset, e.g., estimating the effect of the background color or pen color on the digit value, allowing complete control of the causal effects and uncertainty quantification by Monte Carlo simulations. We train a convolutional neural network on an experiment and we test PPCI on four similar observational studies, varying the treatment effect mechanism and experimental settings distribution, challenging the full support assumption, i.e., Equation 8. A full description of the data-generating processes and analysis is reported in Appendix C. The results are reported in Table 1, clearly showing the DERM potential to solve unsolvable instances for both ERM and invariant training (see soft shifts columns), while suffering when challenging its assumptions (see hard shifts columns).

Table 1: PPCI results on CausalMNIST varying the target experiment population (in-, out-of-distribution), the treatment effect mechanism (linear, non-linear) and challenging the full-support guarantees (soft, hard shifts). We report the ATE estimation bias and standard deviation over 50 repetitions of the data generating process.

| Method | In-Distribution | OoD (*linear effect*) | | OoD (*non-linear effect*) | |
|---|---|---|---|---|---|
| | | Soft shift | Hard shift | Soft shift | Hard shift |
| Ground Truth[3] | $0.00 \pm 0.02$ | $0.00 \pm 0.02$ | $0.00 \pm 0.02$ | $0.04 \pm 0.05$ | $0.03 \pm 0.05$ |
| ERM | $\mathbf{0.00 \pm 0.02}$ | $0.86 \pm 0.14$ | $1.05 \pm 0.15$ | $0.64 \pm 0.18$ | $0.85 \pm 0.17$ |
| vREx | $0.01 \pm 0.03$ | $0.83 \pm 0.15$ | $1.05 \pm 0.14$ | $0.55 \pm 0.17$ | $0.82 \pm 0.14$ |
| IRM | $0.01 \pm 0.03$ | $0.76 \pm 0.18$ | $1.02 \pm 0.12$ | $0.45 \pm 0.18$ | $0.77 \pm 0.17$ |
| DERM (ours) | $0.10 \pm 0.07$ | $\mathbf{0.14 \pm 0.14}$ | $\mathbf{0.75 \pm 0.05}$ | $\mathbf{0.08 \pm 0.27}$ | $\mathbf{0.45 \pm 0.12}$ |

# 6  Conclusion

In this paper, we formalize the problem of obtaining valid causal inference when the outcome variable is latent, and captured by complex measurements. Motivated by scientific applications, we discuss how to train an outcome predictor on similar annotated data, e.g., collected in prior experiments, and achieve valid causal conclusions on the prediction-powered target experiment. We developed the first identification and estimation results for the problem, alongside a practical constraint on the model representation to transfer causal validity under standard yet idealized assumptions. We additionally offered a novel and effective implementation of the such constraint (Deconfounded Empirical Risk Minimization), and showed that it achieves zero-shot identification of the ATE on ISTAnt by leveraging (our) *similar* annotated ecological experiment. Overall, our work offers a paradigm shift from the traditional Causal Representation Learning literature [Schölkopf et al., 2021], paving the way toward learning representations that power downstream causal estimates on real-world data. We think that this has the potential to play a critical role for deep learning models to accelerate the analysis of scientific data and ultimately new discoveries.

The main limitation of this work is that, in practice, it is not possible to verify whether either the conditional calibration property or the representation constraint actually holds on the target population without annotations. Beyond our key assumption that the representation transfers, we also did not discuss the model convergence on finite training datasets, so even if enforcing the representation constraint on training would transfer, we may still suffer from finite sample effects. Another limitation is neglecting several settings that are, potentially, practically relevant, e.g., not overlapping effect modifiers among similar experiments, or the violation of Equation 8. As our target experiments did not have these problems, we left them for future work. Finally, we rest on the assumption that the underlying causal inference problem is statistically identifiable, but this is necessary and can be ensured with a well-designed experimental protocol. Despite these limitations, we hope our work can inspire a new generation of practical Causal Representation Learning methodologies and create opportunities for more systematic and practically relevant benchmarking.

---

[3]By AIPW asymptotic normality on the true factual outcomes.

## Acknowledgments

We thank the Causal Learning and Artificial Intelligence group at ISTA for the continuous feedback on the project and valuable discussions. We thank the Social Immunity group at ISTA, particularly Jinook Oh, for the annotation program and Michaela Hoenigsberger for supporting our ecological experiment. Riccardo Cadei is supported by a Google Research Scholar Award and a Google Initiated Gift to Francesco Locatello. This research was funded in part by the Austrian Science Fund (FWF) 10.55776/COE12). It was further partially supported by the ISTA Interdisciplinary Project Committee for the collaborative project "ALED" between Francesco Locatello and Sylvia Cremer. For open access purposes, the author has applied a CC BY public copyright license to any author accepted manuscript version arising from this submission.

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

Table 1, fully describing valid confidence intervals by sample mean asymptotic normality (Monte Carlo simulation).

Guidelines:

- The answer NA means that the paper does not include experiments.
- The authors should answer "Yes" if the results are accompanied by error bars, confidence intervals, or statistical significance tests, at least for the experiments that support the main claims of the paper.
- The factors of variability that the error bars are capturing should be clearly stated (for example, train/test split, initialization, random drawing of some parameter, or overall run with given experimental conditions).
- The method for calculating the error bars should be explained (closed form formula, call to a library function, bootstrap, etc.)
- The assumptions made should be given (e.g., Normally distributed errors).
- It should be clear whether the error bar is the standard deviation or the standard error of the mean.
- It is OK to report 1-sigma error bars, but one should state it. The authors should preferably report a 2-sigma error bar than state that they have a 96% CI, if the hypothesis of Normality of errors is not verified.
- For asymmetric distributions, the authors should be careful not to show in tables or figures symmetric error bars that would yield results that are out of range (e.g. negative error rates).
- If error bars are reported in tables or plots, The authors should explain in the text how they were calculated and reference the corresponding figures or tables in the text.

8. **Experiments compute resources**

Question: For each experiment, does the paper provide sufficient information on the computer resources (type of compute workers, memory, time of execution) needed to reproduce the experiments?

Answer: [Yes]

Justification: Computer resource requirements are discussed in Appendices B-C.

Guidelines:

- The answer NA means that the paper does not include experiments.
- The paper should indicate the type of compute workers CPU or GPU, internal cluster, or cloud provider, including relevant memory and storage.
- The paper should provide the amount of compute required for each of the individual experimental runs as well as estimate the total compute.
- The paper should disclose whether the full research project required more compute than the experiments reported in the paper (e.g., preliminary or failed experiments that didn't make it into the paper).

9. **Code of ethics**

Question: Does the research conducted in the paper conform, in every respect, with the NeurIPS Code of Ethics https://neurips.cc/public/EthicsGuidelines?

Answer: [Yes]

Justification: Data collection and all experimental work were performed in compliance with international, national and institutional regulations and ethical guidelines. Beyond the new dataset, the authors carefully reviewed the NeurIPS Code of Ethics and believe none of the concerns from NeurIPS Code of Ethics applies to this work.

Guidelines:

- The answer NA means that the authors have not reviewed the NeurIPS Code of Ethics.
- If the authors answer No, they should explain the special circumstances that require a deviation from the Code of Ethics.
- The authors should make sure to preserve anonymity (e.g., if there is a special consideration due to laws or regulations in their jurisdiction).

10. **Broader impacts**

    Question: Does the paper discuss both potential positive societal impacts and negative societal impacts of the work performed?

    Answer: [No]

    Justification: We believe this work has no specific positive or negative societal impacts except to potentially accelerate science on an important topic for biodiversity (which is discussed in the paper).

    Guidelines:

    - The answer NA means that there is no societal impact of the work performed.
    - If the authors answer NA or No, they should explain why their work has no societal impact or why the paper does not address societal impact.
    - Examples of negative societal impacts include potential malicious or unintended uses (e.g., disinformation, generating fake profiles, surveillance), fairness considerations (e.g., deployment of technologies that could make decisions that unfairly impact specific groups), privacy considerations, and security considerations.
    - The conference expects that many papers will be foundational research and not tied to particular applications, let alone deployments. However, if there is a direct path to any negative applications, the authors should point it out. For example, it is legitimate to point out that an improvement in the quality of generative models could be used to generate deepfakes for disinformation. On the other hand, it is not needed to point out that a generic algorithm for optimizing neural networks could enable people to train models that generate Deepfakes faster.
    - The authors should consider possible harms that could arise when the technology is being used as intended and functioning correctly, harms that could arise when the technology is being used as intended but gives incorrect results, and harms following from (intentional or unintentional) misuse of the technology.
    - If there are negative societal impacts, the authors could also discuss possible mitigation strategies (e.g., gated release of models, providing defenses in addition to attacks, mechanisms for monitoring misuse, mechanisms to monitor how a system learns from feedback over time, improving the efficiency and accessibility of ML).

11. **Safeguards**

    Question: Does the paper describe safeguards that have been put in place for responsible release of data or models that have a high risk for misuse (e.g., pretrained language models, image generators, or scraped datasets)?

    Answer: [NA]

    Justification: Both data and analysis don't poses such risks.

    Guidelines:

    - The answer NA means that the paper poses no such risks.
    - Released models that have a high risk for misuse or dual-use should be released with necessary safeguards to allow for controlled use of the model, for example by requiring that users adhere to usage guidelines or restrictions to access the model or implementing safety filters.
    - Datasets that have been scraped from the Internet could pose safety risks. The authors should describe how they avoided releasing unsafe images.
    - We recognize that providing effective safeguards is challenging, and many papers do not require this, but we encourage authors to take this into account and make a best faith effort.

12. **Licenses for existing assets**

    Question: Are the creators or original owners of assets (e.g., code, data, models), used in the paper, properly credited and are the license and terms of use explicitly mentioned and properly respected?

    Answer: [Yes]

Justification: We use our proprietary assets and publicly available public assets (pre-trained Hugging Face models and MNIST and ISTAnt dataset), which we correctly acknowledge and cite.

Guidelines:

- The answer NA means that the paper does not use existing assets.
- The authors should cite the original paper that produced the code package or dataset.
- The authors should state which version of the asset is used and, if possible, include a URL.
- The name of the license (e.g., CC-BY 4.0) should be included for each asset.
- For scraped data from a particular source (e.g., website), the copyright and terms of service of that source should be provided.
- If assets are released, the license, copyright information, and terms of use in the package should be provided. For popular datasets, `paperswithcode.com/datasets` has curated licenses for some datasets. Their licensing guide can help determine the license of a dataset.
- For existing datasets that are re-packaged, both the original license and the license of the derived asset (if it has changed) should be provided.
- If this information is not available online, the authors are encouraged to reach out to the asset's creators.

13. **New assets**

    Question: Are new assets introduced in the paper well documented and is the documentation provided alongside the assets?

    Answer: [Yes]

    Justification: Our main new asset is the real-world dataset. A preview is anonymously shared on Figshare at https://figshare.com/s/9a490b6f6eeebd73350b we plan to publicly release (under license) upon acceptance. A full description of the dataset is reported in Section 5 and Appendix B.

    Guidelines:

    - The answer NA means that the paper does not release new assets.
    - Researchers should communicate the details of the dataset/code/model as part of their submissions via structured templates. This includes details about training, license, limitations, etc.
    - The paper should discuss whether and how consent was obtained from people whose asset is used.
    - At submission time, remember to anonymize your assets (if applicable). You can either create an anonymized URL or include an anonymized zip file.

14. **Crowdsourcing and research with human subjects**

    Question: For crowdsourcing experiments and research with human subjects, does the paper include the full text of instructions given to participants and screenshots, if applicable, as well as details about compensation (if any)?

    Answer: [NA]

    Justification: We do not work with human subjects or crowdsourcing.

    Guidelines:

    - The answer NA means that the paper does not involve crowdsourcing nor research with human subjects.
    - Including this information in the supplemental material is fine, but if the main contribution of the paper involves human subjects, then as much detail as possible should be included in the main paper.
    - According to the NeurIPS Code of Ethics, workers involved in data collection, curation, or other labor should be paid at least the minimum wage in the country of the data collector.

15. **Institutional review board (IRB) approvals or equivalent for research with human subjects**

Question: Does the paper describe potential risks incurred by study participants, whether such risks were disclosed to the subjects, and whether Institutional Review Board (IRB) approvals (or an equivalent approval/review based on the requirements of your country or institution) were obtained?

Answer: [NA]

Justification: We do not work with human subjects or crowdsourcing.

Guidelines:

- The answer NA means that the paper does not involve crowdsourcing nor research with human subjects.
- Depending on the country in which research is conducted, IRB approval (or equivalent) may be required for any human subjects research. If you obtained IRB approval, you should clearly state this in the paper.
- We recognize that the procedures for this may vary significantly between institutions and locations, and we expect authors to adhere to the NeurIPS Code of Ethics and the guidelines for their institution.
- For initial submissions, do not include any information that would break anonymity (if applicable), such as the institution conducting the review.

16. **Declaration of LLM usage**

Question: Does the paper describe the usage of LLMs if it is an important, original, or non-standard component of the core methods in this research? Note that if the LLM is used only for writing, editing, or formatting purposes and does not impact the core methodology, scientific rigorousness, or originality of the research, declaration is not required.

Answer: [NA]

Justification: The core method development in this research does not involve LLMs as any important, original, or non-standard components.

Guidelines:

- The answer NA means that the core method development in this research does not involve LLMs as any important, original, or non-standard components.
- Please refer to our LLM policy (`https://neurips.cc/Conferences/2025/LLM`) for what should or should not be described.

# Appendix

## Table of Contents

## A  Proofs

*Note: In the rest of the manuscript we use the expression "under standard causal identification assumptions" to refer to the canonical conditions for identifiability in a observational study (and so a randomized controlled trial too) [Rubin, 1974], i.e.,:*

- *Stable Unit Treatment Value Assumption (SUTVA), i.e., not interference and no hidden version of the treatment,*

- *Overlap Assumption, i.e., $0 < \mathbb{E}[T|W = w] < 1$ for all $w \in \mathcal{W}$,*

- *Conditional Exchangeability (or Unconfoundness) Assumption, i.e.,*

$$\mathbb{P}(Y|do(T = t), W = w) = \mathbb{P}(Y|T = t, W = w) \qquad \forall w \in \mathcal{W}, t \in \mathcal{T}.$$

### A.1  Proof of Lemma 2.1

> **Lemma** (Prediction-Powered Identification). *Given a PPCI problem $\mathscr{P}$ with standard causal identification assumptions. If an outcome model $g : \mathcal{X} \to \mathcal{Y}$ is conditionally calibrated with respect to the treatment and a valid adjustment set, then it is (causally) valid for $\mathscr{P}$.*

*Proof.* Given the identification of the causal estimand, the thesis follows directly by the tower rule over a valid adjustment set $\tilde{W}$ and leveraging the conditional calibration:

$$\tau_Y(t) = \mathbb{E}[Y|do(T = t)] = \tag{10}$$

$$= \mathbb{E}_{\tilde{W}}[\mathbb{E}_Y[Y|do(T = t), \tilde{W}]] = \tag{11}$$

$$= \mathbb{E}_{\tilde{W}}[\mathbb{E}_Y[Y|T = t, \tilde{W}]] = \tag{12}$$

$$= \mathbb{E}_{\tilde{W}}[\mathbb{E}_X[g(X)|T = t, \tilde{W}]] = \tau_{g(X)}(t) \qquad \forall t \in \mathcal{T}. \tag{13}$$

$$\square$$

## A.2 Proof of Lemma 2.2

**Lemma** (Prediction-Powered Estimation). *Given a PPCI problem $\mathscr{P}$, with identifiable ATE (given the ground-truth annotations). If an outcome model $g : \mathcal{X} \to \mathcal{Y}$ is conditionally calibrated with respect to the treatment and a valid adjustment set, then the corresponding AIPW estimator over the prediction-powered sample with nuisance function estimators satisfying $\|\hat{\mu} - \mu\| \cdot \|\hat{e} - e\| = o_{\mathbb{P}}(n^{-1/2})$, preserves doubly-robustness and valid asymptotic confidence intervals, i.e.,*

$$\sqrt{n}(\hat{\tau}_{g(X)} - \tau_Y) \to \mathcal{N}(0, V), \tag{14}$$

*where $V$ the asymptotic variance.*

*Proof.* Doubly-Robustness follows directly from the conditional calibration assumption and standard doubly-robust argument for AIPW [Robins et al., 1994]. We then focus here on proving the asymptotically normality for $\tau_Y$ of the estimator:

$$\hat{\tau}_{g(X)} = \frac{1}{n} \sum_{i=1}^{n} \left[ \hat{\mu}_{g(X)}(W_i, 1) - \hat{\mu}_{g(X)}(W_i, 0) + \frac{T_i}{\hat{e}(W_i)} (g(X_i) - \hat{\mu}_{g(X)}(W_i, 1)) \right. $$
$$\left. - \frac{1 - T_i}{1 - \hat{e}(W_i)} (g(X_i) - \hat{\mu}_{g(X)}(W_i, 0)) \right] \tag{15}$$

where $\hat{\mu}_{g(X)}(w, t)$ is an estimator of the true predicted-outcome model $\mu_{g(X)}(w, t) = \mathbb{E}[g(X)|W = w, T = t]$, and $\hat{e}(w)$ is an estimator of the true propensity score $e(w) = \mathbb{P}(T = 1|W = w)$.

Given a generic outcome model $\mu$ and a propensity score $e$, let us define the influence function of the estimator:

$$\phi(O_i; \mu, e, g) = \mu(W_i, 1) - \mu(W_i, 0) + \frac{T_i}{e(W_i)} (g(X_i) - \mu(W_i, 1)) $$
$$- \frac{1 - T_i}{1 - e(W_i)} (g(X_i) - \mu(W_i, 0)), \tag{16}$$

where $O_i = (T_i, W_i, Y_i, X_i)$. Then

$$\hat{\tau}_{g(X)} = \frac{1}{n} \sum_{i=1}^{n} \phi(O_i; \hat{\mu}_{g(X)}, \hat{e}, g). \tag{17}$$

We can rewrite our estimator as:

$$\hat{\tau}_{g(X)} - \tau_Y = \frac{1}{n} \sum_{i=1}^{n} [\phi(O_i; \mu, e, g) - \tau_Y] + \frac{1}{n} \sum_{i=1}^{n} \underbrace{[\phi(O_i; \hat{\mu}_{g(X)}, \hat{e}, g) - \phi(O_i; \mu, e, g)]}_{\Delta_i}, \tag{18}$$

where $\mu(w, t) = \mathbb{E}[Y|W = w, T = t]$ is the true outcome model and by conditional calibration:

$$\mu(w, t) = \mathbb{E}[Y|W = w, T = t] = \mathbb{E}[g(X)|W = w, T = t] \qquad \forall w \in \mathcal{W}, t \in \mathcal{T}. \tag{19}$$

Assuming that the second moment of the random variable $\phi$ is bounded, by a standard central limit theorem argument, the first term satisfies

$$\sqrt{n} \left( \frac{1}{n} \sum_{i=1}^{n} \phi(O_i; \mu, e, g) - \tau_Y \right) \xrightarrow{d} \mathcal{N}(0, \underbrace{\mathbb{E}[\phi^2]}_{V}). \tag{20}$$

It remains to show that the second term multiplied by $\sqrt{n}$ goes to zero in probability, i.e. it is asymptotically negligible. To do so, observe that we can rewrite the second term as

$$\frac{1}{n} \sum_{i=1}^{n} \Delta_i = (\mathbb{P}_n - \mathbb{P})(\Delta_i) + \mathbb{P}(\Delta_i), \tag{21}$$

where $\mathbb{P}$ and $\mathbb{P}_n$ are the true and empirical target measures; $\mathbb{P}(\cdot) = \mathbb{E}[\cdot]$ as it is standard in empirical process theory. Our goal is therefore to show that

$$\underbrace{(\mathbb{P}_n - \mathbb{P})(\Delta_i)}_{T_1} + \underbrace{\mathbb{P}(\Delta_i)}_{T_2} = o_{\mathbb{P}}(n^{-1/2}). \tag{22}$$

**Controlling the term $T_1$**   The first term $T_1$ is easy to control, as it follows directly from the following lemma.

---

**Lemma A.1.** *[Kennedy et al., 2020] Let $\widehat{f}(z)$ be a function estimated from a sample $Z^N = (Z_{n+1}, \ldots, Z_N)$, and let $\mathbb{P}_n$ denote the empirical measure over $(Z_1, \ldots, Z_n)$, which is independent of $Z^N$. Then*

$$(\mathbb{P}_n - \mathbb{P})(\widehat{f} - f) = O_{\mathbb{P}}\left(\frac{\|\widehat{f} - f\|}{\sqrt{n}}\right). \tag{23}$$

---

Since we have from assumptions that $\|\phi(\cdot; \hat{\mu}_{g(X)}, \hat{e}, g) - \phi(\cdot; \mu, e, g)\|_2^2 = o_{\mathbb{P}}(1)$, it holds that $T_1 = o_{\mathbb{P}}(n^{-1/2})$.

**Controlling the term $T_2$**   The second term requires some care. We will focus on the term involving $T_i = 1$; the case for $T_i = 0$ follows by symmetry. For $T_i = 1$, after some simple calculations, we have:

$$\Delta_i = \left(\hat{\mu}_{g(X)}(W_i, 1) - \mu(W_i, 1)\right) + \frac{T_i}{\hat{e}(W_i)}\left(g(X_i) - \hat{\mu}(W_i, 1)\right)$$
$$- \frac{T_i}{e(W_i)}\left(g(X_i) - \mu(W_i, 1)\right). \tag{24}$$

Note that we can drop the last term since, by assumption, $g$ and $\mu$ are equal on average. Therefore, we can write:

$$\mathbb{E}[\Delta_i] = \mathbb{E}[\hat{\mu}_{g(X)}(W_i, 1) - \mu(W_i, 1) + \frac{1}{\hat{e}(W_i)}\left(g(X_i) - \hat{\mu}(W_i, 1)\right)] \tag{25}$$

By conditional calibration, we can substitute $g(X_i)$ with $\mu(W_i, 1)$ and group:

$$\mathbb{E}[\Delta_i] = \mathbb{E}\left[\left(\hat{\mu}_{g(X)}(W_i, 1) - \mu(W_i, 1)\right) + \frac{T_i}{\hat{e}(W_i)}\left(\mu(W_i, 1) - \hat{\mu}_{g(X)}(W_i, 1)\right)\right] \tag{26}$$

$$= \mathbb{E}\left[\left(\hat{\mu}_{g(X)}(W, 1) - \mu(W, 1)\right)\left(1 - \frac{T_i}{\hat{e}(W)}\right)\right] \tag{27}$$

$$= \mathbb{E}\left[\left(\hat{\mu}_{g(X)}(W, 1) - \mu(W, 1)\right)\frac{\hat{e}(W) - T_i}{\hat{e}(W)}\right]. \tag{28}$$

Conditioning on $W_i$ we obtain:

$$\mathbb{E}[\Delta_i] = \mathbb{E}\left[\left(\frac{e(W)}{\hat{e}(W)} - 1\right)\left(\mu(W, 1) - \hat{\mu}_{g(X)}(W, 1)\right)\right]. \tag{29}$$

To bound this term, we use the positivity assumption that $\hat{e}(W) \geq \epsilon > 0$ for some constant $\epsilon$:

$$|\mathbb{E}[\Delta_i]| \leq \mathbb{E}\left[\left|\frac{e(W) - \hat{e}(W)}{\hat{e}(W)}\right| \cdot \left|\mu(W, 1) - \hat{\mu}_{g(X)}(W, 1)\right|\right] \tag{30}$$

$$\leq \frac{1}{\epsilon}\mathbb{E}\left[|e(W) - \hat{e}(W)| \cdot |\mu(W, 1) - \hat{\mu}_{g(X)}(W, 1)|\right]. \tag{31}$$

Applying Cauchy-Schwarz inequality:

$$\frac{1}{\epsilon}\mathbb{E}\left[|e(W) - \hat{e}(W)| \cdot |\mu(W, 1) - \hat{\mu}_{g(X)}(W, 1)|\right] \leq \frac{1}{\epsilon}\|e - \hat{e}\|_2\|\mu(\cdot, 1) - \hat{\mu}_{g(X)}(\cdot, 1)\|_2. \tag{32}$$

If the estimators $\hat{e}$ and $\hat{\mu}$ achieve suitable convergence rates such that their product of $L_2$ norms is $o_{\mathbb{P}}(n^{-1/2})$, then:

$$\frac{1}{\epsilon}\|e - \hat{e}\|_2\|\mu(\cdot, 1) - \hat{\mu}_{g(X)}(\cdot, 1)\|_2 = o_{\mathbb{P}}(n^{-1/2}). \tag{33}$$

This completes the proof. $\qquad\qquad\qquad\qquad\qquad\qquad\qquad\qquad\qquad\qquad\qquad\qquad\quad$ $\square$

## A.3   Proof of Theorem 3.1

**Theorem** (Causal Lifting transfers causal validity). *Given two similar PPCI problems $\mathscr{P}$ and $\mathscr{P}'$, with identifiable ATEs given the ground-truth annotations. Let $g = h \circ \phi : \mathcal{X} \to \mathcal{Y}$ be an outcome model for $\mathscr{P}'$ with $h$ Bayes-optimal, and assume that the representation transfers, i.e., $\mathbb{P}(\phi(X)|Y) = \mathbb{P}'(\phi(X)|Y)$ for all $y \in \{\mathbb{P}(Y = y) > 0 \vee \mathbb{P}'(Y = y) > 0\}$. Then the Causal Lifting constraint on $\mathscr{P}$ implies $g$ is valid on $\mathscr{P}$.*

*Intuition*: *The theorem shows that Causal Lifting, supported by other transferability assumptions, guarantees to transfer validity among PPCI problems, regardless of potential shifts in the joint distribution of $(Z, Y)$, e.g., different treatment effect. The core of the proof relies on establishing a hard representation transferability, i.e., $\mathbb{P}(\phi(X)|Y = y, Z = z) = \mathbb{P}'(\phi(X)|Y = y, Z = z)$, given by the assumed representation transferability (on standalone $Y$) and the causal lifting constraint. This is done within the expectation of the conditional calibration, which we have already shown implies validity. Note that, indeed, the (marginal) representation transferability, conditioning over $Y$ alone, does not guarantee transferability conditioning over $Z$ too, potentially due by systematic biases within the encoder (e.g., fully retrieving the outcome signal for a certain subgroup and partially missing it for others).*

*Proof.* First, let us remark that if $\mathscr{P} = \mathscr{P}'$, then Bayes optimality is sufficient for validity. Therefore, we now only focus on the true generalization setting with $\mathscr{P} \neq \mathscr{P}'$. By Lemma 2.1 it is sufficient to show conditional calibration to show the outcome model causal validity. Then, by linearity of the expected value we aim to show:

$$\mathbb{E}_Y[Y \mid Z] \overset{a.s.}{=} \mathbb{E}_X[h(\phi(X)) \mid Z], \tag{34}$$

where $Z = [T, \tilde{W}]$, and $\tilde{W}$ is a valid adjustment set. By the tower rule, we can expand the RHS:

$$\mathbb{E}_X[h(\phi(X)) \mid Z] \overset{a.s.}{=} \mathbb{E}_Y \left[ \underbrace{\mathbb{E}_X[h(\phi(X)) \mid Y, Z]}_{\zeta(Y,Z)} \mid Z \right]. \tag{35}$$

By assumptions $\forall y \in \mathcal{Y}, z \in \mathcal{Z}$:

$$\zeta(y, z) := \mathbb{E}_X[h(\phi(X)) \mid Y = y, Z = z] = \tag{36}$$
$$= \mathbb{E}_X[h(\phi(X)) \mid Y = y] = \qquad (Casual\ Lifting\ constraint) \tag{37}$$
$$= \mathbb{E}'_X[h(\phi(X)) \mid Y = y] = \qquad (representation\ transfers) \tag{38}$$
$$= \mathbb{E}'_X[\mathbb{E}'_Y[Y|\phi(X)] \mid Y = y] = \qquad (Bayes\text{-}optimal\ predictor) \tag{39}$$
$$= \mathbb{E}_X[\underbrace{\mathbb{E}'_Y[Y|\phi(X)]}_{\text{function of}\,\phi(X)} \mid Y = y] = \qquad (representation\ transfers) \tag{40}$$
$$= \mathbb{E}_X[\mathbb{E}'_Y[Y|X] \mid Y = y] = \qquad (sufficiency) \tag{41}$$
$$= \mathbb{E}_X[\mathbb{E}_Y[Y|X] \mid Y = y] = \qquad (similarity) \tag{42}$$
$$= y \qquad (law\ of\ iterated\ expectations) \tag{43}$$

Where with the expected value with superscript prime we refer to the expected value over the data distribution of the problem $\mathscr{P}'$. Substituting back $\zeta(Y, Z)$ we have the thesis. $\qquad\square$

# B   ISTAnt

In this Section, we describe in detail our procedure to test and achieve valid causal inferences on ISTAnt without human annotations but only complex measurements (video), relying on fine-tuning a foundational model on our similar annotated experiment.

## B.1   Training annotated experiment and data recording (ours)

We run an experiment very much like the ISTAnt experiment with triplets of worker ants (one treated focal ant, and two nestmate ants), following the step-by-step design described in their Appendix C [Cadei et al., 2024]. We recorded 5 batches of 9 simultaneously run replicates with similar background pen marking for the dish palettes positioning, producing 45 original videos, of which one had to be excluded for experimental problems, leaving 44 analyzable videos. Then, for each video, grooming events from the nestmate ants to the focal ant were annotated by a single domain expert, and we focus on the 'or' events, i.e., "Is one of the nestmate ants grooming the focal one?", considering up to two possible behavior change per second (similarly for ISTAnt). We used a comparable experimental setup (i.e., camera set-up, random treatment assignment, etc.) except for the following, guaranteeing invariant annotation mechanism, i.e., *similar* experiment.

- **Treatments:** Whereas ISTAnt used two micro-particle applications, our experimental treatments also constitute micro-particle application in two different treatments ($n = 15$ each), but also one treatment completely free of micro-particles (control, $n = 14$), all applied to the focal ant. The three treatments of the ants are visually indistinguishable, independent of micro-particle application.

- **Light conditions:** We created a lower-quality illumination of the nests by implementing a ring of light around the experiment container, resulting in more inhomogeneous lighting and a high-lux ("cold") light effect, compared to the light diffusion by a milky plexiglass sheet proposed in the original experiment. Also, our ant nests had a higher rim from the focal plane where the ants were placed, causing some obscuring of ant observation along the walls. See a comparison of the filming set-up and an example of the resulting recording in Figure 3. We also considered a slightly lower resolution, i.e., 700x700 pixels.

- **Longer Videos**: Whereas ISTAnt annotated 10-minute-long videos, we here annotated 30-minute-long videos, even if the ant grooming activity generally decreases with time from the first exposure to a new environment. Our videos were recorded at 30fps and analysed at 2fps, totaling 158 400 annotated frames in the 44 videos.

- **Other potential distribution shifts**: Other sources of variations from the original experiment are:
    - Whereas ISTAnt used orange and blue color dots, we used yellow and blue.
    - Whereas in ISTAnt, grooming presence or absence was annotated for each frame, we here annotated a single grooming event even if the ant stopped grooming for up to one second but then kept grooming after that, with no other behaviors being performed in between. This means that intermediate frames between grooming frames were also annotated as grooming despite the ant pausing its behavior. Such less exact grooming annotations are faster to perform for the human annotator.
    - The person performing annotation in this experiment was different from the annotators in the ISTAnt dataset, leading to some possible observer effects.

Let's observe that our work models the general pipeline in ecological experiments, where multiple experiment variants are recorded over time, e.g., upgrading the data acquisition technique, and we aim to generalize from a lower to higher quality *similar* experiment.

## B.2   Training and analyses details

We considered the full dataset so recorded for finetuning a pre-trained Vision Transformer, i.e., ViT-B [Dosovitskiy et al., 2020], ViT-L [Zhai et al., 2023], CLIP-ViT-B,-L [Radford et al., 2021], DINOv2 [Oquab et al., 2023], as proposed by Cadei et al. [2024] and we left ISTAnt for testing causal estimation performances relying on artificial predictions. For each pre-trained encoder, we fine-tuned

a multi-layer perception head (2 hidden layers with 256 nodes each and ReLU activation) on top of its *class* token via Adam optimizer ($\beta_1 = 0.9, \beta_2 = 0.9, \epsilon = 10^{-8}$) for ERM, vREx (finetuning the invariance constraint in $\{0.01, 0.1, 1, 10\}$) and DERM (ours) for 15 epochs and batch size 256. So, we fine-tuned the learning rates in $[0.0005, 0.5]$, selecting the best-performing hyper-parameters for each model-method, minimizing the Treatment Effect Bias on the training sample, while guaranteeing good predictive performances, i.e., accuracy greater than 0.8, on a small validation set (1 000 random frames). We computed the ATE at the video level (aggregating the predictions per frame) via the AIPW estimator. We used XGBoost for the model outcome and estimated the propensity score via sample mean (constant) since the treatment assignments are randomized, i.e., RCT. For the outcome model, we consider the following experiment settings for controlling: experiment day, time of the day, batch, position in the batch, and annotator. We run all the analyses using 48GB of RAM, 20 CPU cores, and a single node GPU (`NVIDIA GeForce RTX2080Ti`). The main bottleneck in the analysis is the feature extraction from the pre-trained Vision Transformers. We estimate 72 GPU hours to run the full analysis, despite given a candidate outcome model $g$ already finetuned the standalone prediction-powered causal inference component, excluding feature extraction on the target experiment takes less than a GPU minute.

## C   CausalMNIST

To exhaustively validate our method, we replicated the comparison between DERM (our) enforcing the Causal Lifting constraint and ERM (baseline), vREx and IRM (invariant trainings) in a controlled setting, manipulating the MNIST dataset with coloring, allowing us to (i) cheaply replicate fictitious experiments several times, bootstrapping confidence intervals, and (ii) control the underlying causal effects (only empirically estimated in real-world experiments).

### C.1   Data Generating Process

We considered the following training data distribution $\mathbb{P}^A$:

$$W = Be(0.5) \tag{44}$$
$$U = Be(0.02) \tag{45}$$
$$T = Be(0.5) \tag{46}$$
$$Y = W \cdot \text{Unif}(\{0,1,2,3\}) + T \cdot \text{Unif}(\{0,1,2,3\}) + U \cdot \text{Unif}(\{0,1,2,3\}) \tag{47}$$
$$X := f_X(T, W, Y, U, n_X) \tag{48}$$

representing a Randomized Controlled Trial where $f_X$ is a deterministic manipulation of a random digit image $n_X$ from MNIST dataset enforcing the background color $W$ (red or green) and pen color $T$ (black or white) and padding size $U$ (0 or 8). By exchangeability assumption:

$$\begin{aligned} \text{ATE} &= \mathbb{E}[Y|do(T=1)] - \mathbb{E}[Y|do(T=1)] = \\ &= \mathbb{E}[Y|T=1] - \mathbb{E}[Y|T=0] \\ &= 1.5 \end{aligned} \tag{49}$$

Six examples of colored handwritten digits from CausalMNIST are reported in Figure 4.

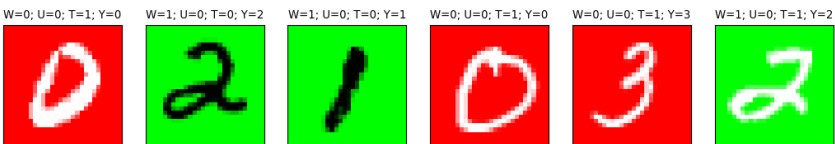

Figure 4: Random samples from a CausalMNIST sample.

Then, we considered the generic *similar* target Observational Study, with distribution:

$$W = Be(p_W) \tag{50}$$
$$U = Be(p_U) \tag{51}$$
$$T = Be(0.1) \cdot (1 - W) + Be(0.9) \cdot W \tag{52}$$
$$X := f_X(T, W, Y, U, n_X) \tag{53}$$

with *linear* outcome/effect (null):

$$Y = W \cdot \text{Unif}(\{0,1,2,3\}) + \text{Unif}(\{0,1,2,3\}) + U \cdot \text{Unif}(\{0,1,2,3\}) \tag{54}$$

where simply:

$$\text{ATE} = \mathbb{E}[Y|do(T=1)] - \mathbb{E}[Y|do(T=1)] = 0 \tag{55}$$

and *non-linear* outcome/effect:

$$Y = (T \vee U) \cdot \text{Unif}(\{0,1,2,3\}) + \text{Unif}(\{0,1,2,3,4,5,6\}) \tag{56}$$

with respect to the experimental settings, where, by adjustment formula:

$$\begin{aligned} \text{ATE} &= \mathbb{E}[Y|do(T=1)] - \mathbb{E}[Y|do(T=1)] = \\ &= P(U=0) \cdot (\mathbb{E}[Y|T=1, U=0] - \mathbb{E}[Y|T=0, U=0]) = \\ &= (1 - p_U) \cdot 1.5 \end{aligned} \tag{57}$$

and we considered 4 different instances $\mathbb{P}^B$, $\mathbb{P}^C$, $\mathbb{P}^D$ and $\mathbb{P}^E$ varying the outcome model and experimental settings parameters. $\mathbb{P}^B$ and $\mathbb{P}^C$ have null effect, while $\mathbb{P}^D$ and $\mathbb{P}^E$ have heterogeneous effect. $\mathbb{P}^B$ and $\mathbb{P}^D$ are closer in distribution to the training since $p_U \ll 1$, while $\mathbb{P}^C$ and $\mathbb{P}^E$ are more out-of distribution since the padding variable $U$ is balanced, i.e., $p_U = 0.5$, but during training it is rarely observed activated. For simplicity we refer to $\mathbb{P}^B$ as the distribution of the the target experiment population with linear effect and "soft" (experimental settings) shift, $\mathbb{P}^C$ with linear effect and "hard" shift, $\mathbb{P}^D$ with non-linear effect and "soft" shift, $\mathbb{P}^D$ with non-linear effect and "hard" shift. In Table 2 we summarize the different distribution parameters, also reporting the corresponding (exact) ATE value.

Table 2: Summary of the training and target experiment distributions.

|  | In-Distribution | OoD (*linear effect*) | | OoD (*non-linear effect*) | |
|---|---|---|---|---|---|
|  |  | Soft shift | Hard shift | Soft shift | Hard shift |
| Distribution | $\mathbb{P}^A$ | $\mathbb{P}^B$ | $\mathbb{P}^C$ | $\mathbb{P}^D$ | $\mathbb{P}^E$ |
| $p_W$ | 0.5 | 0.05 | 0.5 | 0.2 | 0.5 |
| $p_U$ | 0.02 | 0.05 | 0.5 | 0.2 | 0.5 |
| Randomized | True | False | False | False | False |
| Effect | Linear | Null | Null | Non Linear | Non Linear |
| ATE | 1.5 | 0 | 0 | 1.2 | 0.75 |

These data generating processes represent fictitious experiments where we aim to quantify the effect of the pen color on the number to draw (if asked to pick one), relying on a machine learning model for handwritten-digits classification. Particularly, the shift in the treatment effect between training (ATE= 1.5) and target population (ATE$\in \{0, 0.75, 1.2\}$), may be interpreted as (i) testing a different, still homogeneous group of individuals, or (ii) varying the effect by changing the brand of the pens, unobserved variable (perfectly retrievable by the pen color on the training distribution), while keeping the same colors, reflecting the crucial challenges described in Section 2.1.

## C.2 Additional Ablations: Causal Estimators

In addition to the main experiments on CausalMNIST we repeated all the experiments fine-tuning the predictive model via ERM and DERM, and varying the causal estimators. We particularly considered AIPW, X-Learner [Künzel et al., 2019], BART [Chipman et al., 2010], and Causal Forest [Wager and Athey, 2018]. Results are reported in Table 3. As expected, the same biases observed with AIPW are consistently confirmed by all the other estimators, confirming how biases in the prediction-powered samples invalidate any downstream analysis, regardless of the (causal) estimator [Cadei et al., 2024, Cadei and Internò, 2025].

Table 3: ATE bias with standard deviation over 50 repetitions on prediction-powered samples varying the causal estimators. Consistent biases are confirmed by different treatment effect estimators over the same prediction-powered sample.

| Method | Estimator | In-Distribution | OoD (*linear effect*) | | OoD (*non-linear effect*) | |
|---|---|---|---|---|---|---|
|  |  |  | Soft shift | Hard shift | Soft shift | Hard shift |
| ERM | AIPW | **0.00 ± 0.02** | 0.86 ± 0.14 | 1.05 ± 0.15 | 0.64 ± 0.18 | 0.85 ± 0.17 |
|  | X-Learner | 0.01 ± 0.02 | 0.82 ± 0.16 | 0.92 ± 0.15 | 0.68 ± 0.19 | 0.79 ± 0.18 |
|  | BART | 0.01 ± 0.02 | 0.81 ± 0.19 | 0.89 ± 0.13 | 0.59 ± 0.21 | 0.75 ± 0.16 |
|  | Causal Forest | 0.01 ± 0.02 | 0.82 ± 0.13 | 0.90 ± 0.11 | 0.64 ± 0.16 | 0.76 ± 0.12 |
| DERM | AIPW | 0.10 ± 0.07 | 0.14 ± 0.14 | 0.75 ± 0.05 | 0.08 ± 0.27 | **0.45 ± 0.12** |
|  | X-Learner | 0.32 ± 0.27 | **0.00 ± 0.25** | 0.69 ± 0.12 | 0.05 ± 0.27 | 0.53 ± 0.14 |
|  | BART | 0.19 ± 0.14 | 0.07 ± 0.14 | 0.68 ± 0.15 | 0.06 ± 0.18 | 0.52 ± 0.18 |
|  | Causal Forest | 0.22 ± 0.19 | 0.11 ± 0.13 | **0.67 ± 0.09** | **0.03 ± 0.24** | 0.50 ± 0.15 |

### C.3 Training and analyses details

We sampled 10 000 observations from $\mathbb{P}^A$ to train a digits classifier (a Convolutional Neural Network) and tested it in PPCI in-distribution (10 000 more sample from $\mathbb{P}^A$) and out-of-distribution (zero-shot) on 10 000 obervations for each $\mathbb{P}^B$, $\mathbb{P}^C$, $\mathbb{P}^D$, $\mathbb{P}^E$. We replicated the modeling choices for CausalMNIST proposed in Cadei et al. [2024] and described in their Appendix E.2 (without relying on pre-trained models). Particularly, the proposed network consists of two convolutional layers followed by two fully connected layers. The first convolutional layer applies 20 filters of size 5x5 with ReLU activation, followed by a 2x2 max-pooling layer. The second convolutional layer applies 50 filters of size 5x5 with ReLU activation, followed by another 2x2 max-pooling layer. The output feature maps are flattened and passed to a fully connected layer with 500 neurons and ReLU activation. The final fully connected layer reduces the output to ten logits (one per digit) on which we apply a softmax activation to model the probabilities directly. Table 4 reports a full description of the training details for such network. Particularly we tuned the number of epochs in $\{10, 11, ..., 50\}$ and learning rate in $\{0.01, 0.001, 0.0001, 0.00001\}$ by minimizing the Mean Squared Error on a small validation set (1 000 images) and finally retraining the model on the full training sample with the optimal parameters.

Table 4: Training details for the Convolutaional Neural Network training on CausalMNIST.

| Hyper-parameters | Value |
| --- | --- |
| Loss | Cross Entropy |
| Learning Rate | 0.0001 |
| Optimizer | Adam ($\beta_1 = 0.9, \beta_2 = 0.9, \epsilon = 10^{-8}$) |
| Batch Size | 32 |
| Epochs | 40 |

For each learning objective, i.e., DERM (ours), ERM, vREx, and IRM, we trained a model on the training sample, imputed the outcome in each target sample, and estimated the ATE via a causal estimator (AIPW for main experiments, AIPW, X-Learner, Causal Forest, and BART for the additional experiments). Within the AIPW and X-Learner estimator, we used the XGBoost regressor for the outcome regression and the XGBoost classifier to estimate the propensity score on observational studies (or vanilla sample mean on randomized controlled trials since constant). We rely on default parameters for both Causal Forest and BART (with 10 trees) as suggested in the original papers. For both IRM and vREx, we replicated the same hyper-parameter tuning of the invariant coefficient from our experiments on ISTAnt generalization, selecting in both cases the invariant coefficient $\lambda = 0.1$. We repeated each experiment 50 times, including resampling the data, and bootstrapped the confidence interval of the ATE estimates. We run all the analysis using 10GB of RAM, 8 CPU cores, and a single node GPU (`NVIDIA GeForce RTX2080Ti`). The main bottleneck of each experiment is re-generating a new version of CausalMNIST from MNIST dataset, and then training the model. The Prediction-Powered ATE estimation is significantly faster. We estimate a total of 12 GPU hours to reproduce all the experiments described in this section.

