# OpenReview forum: "Prediction-Powered Causal Inferences"
_NeurIPS.cc/2025/Conference — NeurIPS 2025 poster_

### Official Review · Reviewer_N6pk · 2025-06-20

**Clarity:** 1
**Significance:** 3
**Originality:** 3
**Rating:** 4
**Confidence:** 4

**Summary:**

This paper proposes an idea of estimating causal effects from high dimensional observation without labels. Then the paper discusses the desiderata of a model to estimate causal effects. Based on the analysis, the paper proposes a method called deconfounded empirical risk minimization and shows improved performance over several baseline methods for estimation causal effects.

**Questions:**

Please see the weaknesses section and address those points, especially the points on the validity of assumptions and the underlying causal model.

**Ethical Concerns:**

["NO or VERY MINOR ethics concerns only"]

**Final Justification:**

Assuming that the authors will improve the paper's presentation and readability in the next revision, I am leaning towards positive score.

**Limitations:**

Yes.

**Paper Formatting Concerns:**

None.

**Quality:**

3

**Strengths And Weaknesses:**

Strengths:

The idea of estimating causal effects from high dimensional measurements without explicit label for the observed data is novel and helps the community to solve the causal effect estimation task innovatively. Experimental results show the promising results of the proposed method compared to baselines.

Weaknesses:

The major weakness of this paper is its readability. I had to read several times to understand the goal of the paper.
1. Here are some issues in the introduction.
      (a) The first paragraph reads like the paper is about to solve an explainability problem.

      (b) In L22, it is not clear what it means to "analyzing experimental data" by a trained model.

      (c) In L29, it is not clear what "outcome predictions" means. outcomes and predictions are usually the same.

      (d) A real-world example in the introduction would greatly help its readability.

2. In section 2, having a causal graph to model the setting and assumptions is required to understand the setting clearly.
3. $\mathcal{I}$ is not defined in L77. When the ISTAnt task is introduced in section 2, it is not clear what the goal of the task is. The full form of AIPW is never presented.
4. Grammatical erros and typos: L97: "identify in a statistical estimated", L305: "convolutiona"
5. Inconsistencies: $W$ in L101 and $\tilde{W}$ in 174 are used to denote the same thing i.e., valid adjustment set.
6. Explaining conditional calibration and causal lifting with real-world examples would help understand the paper better.
7. (Important) The standard causal identification assumptions rarely hold in real world data. For instance, what measures are taken to ensure that there is no hidden/unobserved confounding variables? This will subsequently influence the performance of the proposed deconfounded empirical risk minimization algorithm.

---

> ### Author Rebuttal · Authors · 2025-07-31
>
> We thank the reviewer @N6pk for the feedback. Overall, the general objective and purpose of the paper seem delivered and appreciated (“_novel idea_”, “_promising results_”), even if the raised concerns suggest a **potential misunderstanding** of the **prediction-powered setting** and **related literature**. We clarify here.
>
> **Causal Model Assumption** (Weakness 2)
> - The PPCI formulation does _not require_ introducing a causal model, whose measurement-annotation relationship could also be controversial, as suggested by reviewer **`@7mz6`**. It simply relies on extending a well-defined treatment effect estimation problem (with its local causal assumptions), replacing the ground-truth (factual) outcomes with model predictions from high-dimensional predictive measurements.
>
> **Latent confounding** (Weakness 7)
> - Note that we are *assuming* that the causal estimand is statistically *identifiable*. If that is not the case, then even with ground truth $Y$, the causal estimation is ill-posed and so it doesn’t make much sense to investigate the Prediction-Powered extension of it.
> In our experiment, this is ruled out by the fact that we consider randomized trials, which are one of the many possible ways to guarantee identification (one could have instruments, or feasible backdoor adjustments).
>
> The reviewer raised a number of valuable **minor concerns** and clarification requests that we now address, and we plan to adapt in the final version of the manuscript, hoping to **improve** its **clarity** accordingly.
>
> - *Introduction* (Weakness 1)
>   - The first paragraph in the introduction *doesn’t refer to explainability*, if not for the expression “_black-box_” referring to a model with generally no guarantees as to how it reaches its predictions, especially required in scientific applications. We are happy to make this clearer.
>   - Given the *2-stage* nature of the *problem*, i.e., factual outcome estimation and causal effect estimation, we need to specify the object of prediction.
>   - A real-world problem (zero-shot generalization to ISTAnt) is used as a *motivating example* from the introduction to the conclusion to support problem interpretation, additionally illustrated in Fig. 1-3 and Example 1-2. As suggested, we plan to extend the dataset presentation in the introduction.
> - *Typos* (Weaknesses 3a, 4 , 5).
>   - We will replace $\mathcal{I}$ with $i$ ranging from $1$ to $n$. We thank the reviewer for spotting it, together with the *other typos*.
> - *Task Clarification* (Weakness 3b, 3c)
>   - ISTAnt *task is* explicitly *presented* in L89-90, “_ATE estimation on the ISTAnt dataset zero-shot, i.e., without behaviour annotations. Instead, we leverage our own experiment variant with different treatment and recording platform_” and additionally illustrated in Fig. 1. After extending the example presentation in the introduction, we hope this will be clearer.
>   - AIPW [Robins et al. 1994] is the *gold standard* for ATE estimation and is assumed to be known according to common practices in the field.
> - *Clarification by Real-world Example* (Weakness 6)
>   - The Casual Lifting constraint is *already explained* by our motivating example (see L182-189), while the Conditional Calibration interpretation is self-explicatory, still interpreted several times, e.g., in Example 1 (see L149-152).
>
> [1] J.M. Robins et al, Estimation of regression coefficients when some regressors are not always observed, JASA 1994.

---

> > ### Comment · Reviewer_N6pk · 2025-08-03
> > **Thanks for the response**
> >
> > I thank the authors for the clarifications. In its current form the manuscript’s presentation and writing style make key ideas hard to follow. I hope the authors will improve readability in their next revision.

---

### Official Review · Reviewer_4Lsh · 2025-06-27

**Clarity:** 4
**Significance:** 4
**Originality:** 4
**Rating:** 5
**Confidence:** 4

**Summary:**

This paper addresses the problem of zero-shot causal inference from high-dimensional experimental data—such as images or videos—where ground truth outcomes are unavailable in the target experiment. The authors formalize this setting as Prediction-Powered Causal Inference (PPCI) and study conditions under which a model trained on a different but similar experiment can be used to estimate causal effects, such as the Average Treatment Effect (ATE), without further fine-tuning or annotation.

To support valid zero-shot PPCI, the paper introduces a constraint called causal lifting, which requires the learned representation phi(X) to be independent of experimental settings Z = [T, W] given the outcome Y, i.e., I(phi(X); Z | Y) = 0. The authors propose a simple, model-agnostic implementation of this constraint via Deconfounded Empirical Risk Minimization (DERM), which reweights training data to simulate a distribution where outcome and covariates are uncorrelated.

Empirical results demonstrate that DERM enables valid zero-shot ATE estimation on a new variant of the ISTAnt ecological dataset, where prior methods fail. The authors further validate their method on a controlled synthetic benchmark (CausalMNIST), showing robustness across several types of distribution shifts.

**Questions:**

* Would it help to give a more detailed discussion of when DERM fails, particularly in cases of limited support or latent confounding?
* Would it be possible to connect the causal lifting constraint more directly to existing literature on conditional independence in representation learning, to clarify what is truly new?
* Would it help to give a brief comparison or mention of adversarial methods or information bottleneck approaches for enforcing independence? This might provide useful context.

**Ethical Concerns:**

["NO or VERY MINOR ethics concerns only"]

**Final Justification:**

The authors have made inreads into clarity, though I was already happy with the paper. I keep my score.

**Limitations:**

The authors have mentioned a possible acceleration of automated science as a potential societal side effect. They could also mention the drawback of the work being used inappropriately, for example, when attempting to apply it in a context where latent confounding is a genuine issue.

**Paper Formatting Concerns:**

None.

**Quality:**

4

**Strengths And Weaknesses:**

This paper addresses a practical problem—zero-shot causal effect estimation from high-dimensional observations—by proposing a representation-level constraint (“causal lifting”) and a practical implementation (DERM). The authors demonstrate the approach on a real ecological experiment and a synthetic benchmark, showing improved generalization in settings where standard ERM and invariant training methods fail. The paper is clearly written, well-motivated, and methodologically sound.

Strengths:
* The problem of prediction-powered causal inference (PPCI) is well-framed and highly relevant for scientific domains where annotation is expensive.
* The causal lifting constraint, while not novel in its form, is a well-chosen and effective principle for enforcing robustness.
* The DERM procedure is simple, implementable, and grounded in sound intuitions from the representation learning and fairness literature.
* The procedures deal effectively with scenarios where latent confounding is not an issue, but selection bias may be.
* The experimental results are strong, including a new ISTAnt-like dataset and a successful zero-shot transfer.
* The paper is careful in stating assumptions, and the theoretical claims are well supported.

Weaknesses / Limitations:
* The causal lifting constraint amounts to a conditional independence assumption (i.e., I(\phi(X); Z \mid Y) = 0), which is well-known in prior literature. The framing is helpful, but the novelty is conceptual rather than technical.
* The approach assumes no latent confounding, which may limit its applicability in real-world settings. This assumption is acknowledged but perhaps not emphasized enough.
* DERM depends critically on a full-support condition (i.e., \text{supp}(Y \mid Z=z) = \text{supp}(Y) for all z) to simulate outcome–covariate independence. The failure of this condition can undermine the procedure, and this could be discussed more explicitly.
* The method may be sensitive to variance in the reweighting procedure when sample sizes are small or the covariates are unbalanced.

---

> ### Author Rebuttal · Authors · 2025-07-31
>
> We thank the reviewer @4Lsh for the curated feedback, which summary and strengths/weaknesses show the in-depth understanding and appreciation of the paper. We answer here to the raised questions and plan to integrate such clarifications in the final version of the manuscript, together with the suggested limitations emphasis.
>
> **Causal Lifting constraint: novelty and related works** (Weakness 1, Questions 2-3)
> - In this paper, we characterize which properties a (factual outcome) predictor requires to be incorporated in a causal inference pipeline without training on the target population. Alongside, we also propose a methodology for achieving them, relying on similar data. Particularly, we show that enforcing conditional independence between the measurement representation and the experimental settings, given the outcome, transfers Causal Validity among similar experiments (Thm. 3.1). The novelty lies in such formulation, which implementation happens to be a well-studied problem in Representation Learning for other reasons. We reviewed this in the Invariant Representation paragraphs in Section 4.
> - Methodologically, we chose a specific implementation of the invariance constraint, which is far simpler than adversarial [Louizos et al. 2016] or information bottleneck [Alemi et al. 2017] approaches which are known to be unstable and inefficient respectively. We are happy to expand in Section 4 on the advantages of the different approaches and why we chose DERM for our particular application, see next.
>
>
> **DERM Limitations**: full support and finite sample setting (Weaknesses 3-4, Question 1)
> - We practically designed our ad-hoc implementation, motivated by our application of interest (high-dimensional representation, with discrete and limited covariates), but other approaches could and should be considered depending on the problem characteristics. As recognised in the limitations, DERM has no guarantees beyond the full support assumption (as shown with Hard shifts on CausalMNIST experiments), and new methods, tailored for such a regime, or just a strongly unbalanced distribution, surely deserve to be investigated.
>
> **Latent confounding** (Weakness 2, Question 1)
> - Note that we are assuming that the causal estimand is statistically identifiable. If that is not the case, then even with ground truth $Y$, the causal estimation is ill-posed. In our real-world experiment, this is ruled out by the fact that we consider randomized trials, which are one of the many possible ways to guarantee identification.
>
> [1] Louizos, Christos et al. "The Variational Fair Autoencoder" ICLR 2016
>
> [2] Alemi, Alexander A. et al. "Deep variational information bottleneck" ICLR 2017

---

> > ### Comment · Reviewer_4Lsh · 2025-08-03
> >
> > Thanks for the response. Considering only the issues I raised, I'm satisfied with the proposals.
> >
> > I've noticed that I'm out of step with my fellow reviewers in my assessment, and I hope their concerns can be adequately addressed. In particular, I must admit that I had a similar problem to Reviewer N6pk, as I also had to read the paper several times to understand its content; however, once I grasped the point, I was fairly satisfied with it. But clarity is a real problem and can be addressed. The paper should ideally be accessible to someone not working specifically in this field.

---

> > > ### Author Response · Authors · 2025-08-03
> > >
> > > We thank the reviewer for the positive feedback.
> > >
> > > We acknowledge that our work introduces many new concepts that bridge classical ideas from causal inference and deep learning, requiring us to strike a **balance between precision and accessibility**. This is particularly relevant in light of some misunderstandings around foundational causal concepts, which we believe are natural given the broad NeurIPS’ audience. Nevertheless, *Prediction-Powered Causal Inference* raises important new challenges at the intersection of these fields, and we agree it is crucial to clearly communicate these ideas and actively **engage both communities** in shaping and advancing this direction. We think that these improvements are within the scope of a camera ready revision, and we are committed to improving clarity throughout the paper.

---

### Official Review · Reviewer_pzYf · 2025-07-02

**Clarity:** 2
**Significance:** 3
**Originality:** 3
**Rating:** 3
**Confidence:** 3

**Summary:**

The paper introduces Deconfounded Empirical Risk Minimization (DERM), a weighted extension of Empirical Risk Minimization designed to address confounding in treatment effect estimation when only high-dimensional covariates are available. DERM enables Prediction-Powered Causal Inference (PPCI), estimating treatment effects in unlabeled target experiments using predictions from pre-trained models. The method enforces a new constraint, Causal Lifting, to ensure causal validity transfers zero-shot across experiments. The authors provide both theoretical analysis and practical implementation, showing DERM outperforms standard ERM and invariant training on synthetic and real-world datasets.

**Questions:**

- Is “retrievable zero-shot from a pre-trained model” a predicate for the factual outcomes? What exactly is retrievable, and how?
- What does “this problem” refer to in line 30? Why is it labeled a zero-shot problem before the pretrained model is introduced?
- Can the authors elaborate on the source of bias in an RCT setting mentioned in lines 33–34?
- Would Lemma 2.2 hold for any standard ATE estimator under the given assumptions, not just under \(g\)?
- In the causal setting, would \(X\) be a parent of \(Y\) in a causal graph, or vice versa? Clarification would improve causal interpretation.
- How is DERM defined given that no labels for \(Y\) are available? How are the weights \(w_i\) constructed in this case?
- Is the definition of causal lifting introduced by the authors, or does it extend prior work?
- Could the authors include Conditional Average Treatment Effect (CATE) results, given their relevance to ML applications?
- Would a footnote on valid adjustment sets and identifiability improve clarity and completeness?

**Ethical Concerns:**

["NO or VERY MINOR ethics concerns only"]

**Final Justification:**

The authors addressed my questions. While some parts of the work are well presented, I still believe the paper needs more work to improve its clarity and include further results for the conditional setting to be more complete. Hence, I am keeping my assessment.

**Limitations:**

yes.

**Paper Formatting Concerns:**

None.

**Quality:**

3

**Strengths And Weaknesses:**

## Strengths
- The problem statement in Section 2 is excellent and clearly written.
- The definition of causal lifting is neat, appealing, and well-presented.
- The theoretical analysis, while relatively simple, is presented in a smooth and pleasant manner.
- The empirical results are good, and the general seem to be highly relevant (I am still not getting the exact setting fully though).
- The paper’s structure and intention become clearer as it progresses, particularly in the formal sections.

## Weaknesses
- The paper sometimes contain some overly complicated wording in my honest opinion.
- “Retrievable zero-shot from a pre-trained model” – it's ambiguous whether this refers to factual outcomes.
- “Powering causal inferences” – sounds awkward; “estimating treatment effects” is clearer.
- “The interventional effect of a treatment” – overly complicated; simpler phrasing like “treatment effect” would help.
- Lines 33–35 create confusion in my opinion, if no labels are available, how is training done at all?
- “Conditional calibration” is mentioned multiple times without definition; to my knowledge this is not a standard term and needs at least a brief explanation before formally defining it later.
- Lines 40–47 are dense and unclear. The discussion of domain shift and ERM bias is disconnected from the context of pretrained models.
- Example 1 is abstract and could benefit from a more concrete illustration.
- Example 2 uses vague terminology; more specific examples would clarify the scenario.
- Some definitions (e.g., Definition 2.1) include conditions that may be unnecessary or confusing, such as tying identifiability of $Y$ to a property of $g$.
- The introduction lacks the clarity and precision found in the formal sections, making it harder to follow early on.

---

> ### Author Rebuttal · Authors · 2025-07-31
>
> We thank reviewer @pzYf for the feedback. The methodological contribution of the paper seems properly delivered, while our **problem formulation and theoretical analysis** seems **ignored** as a contribution (no mention in reviewer’s summary and questioned Causal Lifting contribution) and **partially misinterpreted** (Zero-Shot meaning and implementation, Prediction-Powered Estimation Theorem) despite appreciated (“_excellent and clearly written_”, “_neat, appealing, and well-presented_”, “_smooth and pleasant_”, “_highly relevant_”). **No major concern** is raised, and weaknesses mostly revolve around unclear phrasing that we commit to improving. We address the raised questions in order, hoping to clarify potential misinterpretation:
>
> - Q1: _“Is ‘retrievable zero-shot from a pre-trained model’ a predicate for the factual outcomes? What exactly is retrievable, and how?”_
>
>   - Retrievable refers to the factual outcomes estimated by a predictive model. We agree it is not the best wording, and we will replace ‘‘_retrievable_” with “_predicted_”.
>
> - Q2: _“What does ‘this problem’ refer to in line 30? Why is it labeled a zero-shot problem before the pretrained model is introduced?”_
>   - Zero-shot refers to predicting the factual outcomes “_without training on the data from the target experiment_” (L35), transferring knowledge from a pre-trained model and similar training data (see Def. 2.3), while preserving the properties in the downstream causal estimation.
>
> - Q3: _“Can the authors elaborate on the source of bias in an RCT setting mentioned in lines 33–34?”_
>   - Biases in prediction-powered average treatment effect estimation can arise from different reasons, as discussed by Cadei et al. (2024). In L33-34, we refer to Lemma 3.1 from the same paper, describing how small biases in prediction (i.e., $1-\epsilon$ accuracy with systematic errors in a subgroup) can invalidate the downstream causal estimation even in the simplest causal setting, i.e., a Randomized Controlled Trial.
>
> - Q4: _“Would Lemma 2.2 hold for any standard ATE estimator under the given assumptions, not just under (g)?”_
>   - We prove Thm. 2.2 only for AIPW, gold standard for ATE estimation on RCT (e.g., in our experiments on ISTAnt), and observational studies with suitable backdoor adjustment. It shows how the properties of a causal estimator, i.e., AIPW, are preserved by a conditional calibrated factual outcome model $g$. Similar results may be shown for other estimators.
>
>
> - Q5: _“In the causal setting, would (X) be a parent of (Y) in a causal graph, or vice versa? Clarification would improve causal interpretation.”_
>   - For ease of interpretation, we assumed the outcome ($Y$) as a cause of the high-dimensional observation ($X$), see L139. Intuitively, $Y$ is something that happens in the real world, and X is its measurement, e.g., video recording (changing $Y$ affects X, but changing the pixels in $X$ does not affect what happens in the experiment).  However, we also never explicitly rely on such an assumption, but rather on $P(Y|X)$ invariance.
>
> - Q6: _“How is DERM defined given that no labels for (Y) are available? How are the weights (w_i) constructed in this case?”_ and weakness _“lines 33–35 create confusion in my opinion,  if no labels are available, how is training done at all?”_
>   - DERM is the method we propose to enforce the Causal Lifting constraint by training (finetuning) a model on annotated data from similar experiments (see Def. 2.3), generally out-of-distribution for the non-annotated target experiment (by design). It reflects a realistic and common setup for the type of applications we target, e.g., ecologists study ants' behaviors across sequential similar experiments. In practice, it estimates the weights $w_i$ and operates on such annotated similar experiments, attempting to transfer causally valid predictions to the target experiment.
>
> - Q7: _“Is the definition of causal lifting introduced by the authors, or does it extend prior work?”_
>   - Yes, the definition is new, and it answers the challenge raised by Cadei et al. (2024) to estimate the treatment effect on the predictions of pre-trained models. Indeed, without further assumptions, a pre-trained model is generally biased, e.g., under-performing in a specific subgroup, and we introduce the expression Causal Lifting to refer to the procedure of correcting such biases.
>
> - Q8: _“Could the authors include Conditional Average Treatment Effect (CATE) results, given their relevance to ML applications?”_
>   - To the best of our knowledge, ISTAnt by Cadei et al. (2024) is still the unique benchmark for PPCI on high-dimensional measurements, and only evaluates ATE estimation. Through this paper, we have already contributed to it by introducing a similar real experiment for training (simulating a realistic PPCI pipeline). We agree on the relevance of extending PPCI benchmarks to heterogeneous effects, despite being outside the scope of this paper.
>
> - Q9: _“Would a footnote on valid adjustment sets and identifiability improve clarity and completeness?”_
>   - Fair point, we are happy to stress this further. We are indeed assuming that the causal estimand is statistically identifiable. If that is not the case, then even with ground truth Y, the causal estimation is ill-posed. In our experiment, this is ruled out by the fact that we consider randomized trials, which are one of the many possible ways to guarantee identification (one could have instruments, or feasible backdoor adjustments).
> ​​
>
> The reviewer suggested a few **other minor wording improvements**, which are listed as the paper's weaknesses. Here is what we suggest:
>
>   - We will replace “_Retrievable zero-shot_” with “_zero-shot predictions_”.
>   - We will replace “_Powering causal inferences_” with “_prediction powered treatment effect estimation_”, which more precisely follows established Prediction Powered Inference terminology in statistics [Anastasios N. et al. 2023].
>   - We will avoid mentioning “_Conditional Calibration_” before its definition in Definition 2.2, unless to explicitly claim it as a novel contribution and point to the definition.
>   - In Definition 2.1 we assume that the causal estimand is statistically identifiable in $Y$ and define that a predictor for $Y$ is valid only if the causal estimand is also identifiable in $g(X)$. This definition ties the identifiability of $g(X)$ to that of $Y$, not viceversa. We’ll make sure this is clearer in the draft using the footnote on identifiability suggested in Q9.
>   - We thank the reviewer for the other specific suggestions where clarity can be improved. We will take this into account in the final revision and improve the presentation.
>
> [1] Cadei, Riccardo et al. “Smoke and Mirrors in Causal Downstream Tasks” NeurIPS 2024
>
> [2] Angelopoulos, Anastasios N. et al. “Prediction-Powered Inference” Science 2023

---

> > ### Comment · Reviewer_pzYf · 2025-08-04
> >
> > Thank you for your answers.
> >
> > First, I would like to mention that I do not appreciate picking some positive words from my review-that were written specifically about specific parts of the paper- and present them as my general perception of the entire work. I find the way the authors summarized my review highly misleading.
> >
> > Second, I appreciate the response to my questions. I would have appreciated if the authors summarized where the bias is coming from in Q3 (as this has been only studied in one previous paper and is not an established known fact in the literature and that seems to me to be an important motivation for the work.

---

> > > ### Author Response · Authors · 2025-08-04
> > >
> > > We apologize for the **misunderstanding of our intent** in the summary of the positive points. We aimed to highlight that the reviewer found the problem formulation and analysis positive despite potentially **missing** some aspects of our **contributions** (causal lifting framework).  Our main intent was to stress that the paper’s contribution goes beyond introducing a new method (DERM), as it may instead appear from the Reviewer summary. Nevertheless, we apologize for the misunderstanding.
> > >
> > > Concerning the source of biases, we focus on the ones arising from the (pre)training data and the architecture, also discussed by Cadei et al. (2024). Of the other biases they describe, discretization bias is easy to fix, and selection bias is a special case of what we propose (and for which they offered no solution). We offer a new, more precise characterization of the sources of bias that are critical in our setting and were not described before. These are described in Examples 1 and 2 using real-world examples. We will improve the Figures and descriptions thanks to the reviewer’s feedback. Regardless of the sources of biases, however, this work objective concerns their mitigation, which is also crucially missing in Cadei et al. (2024), except for discretization bias, which is trivial to fix.
> > >
> > > [1] Cadei, Riccardo et al. “Smoke and Mirrors in Causal Downstream Tasks” NeurIPS 2024

---

### Official Review · Reviewer_7mz6 · 2025-07-02

**Clarity:** 4
**Significance:** 3
**Originality:** 3
**Rating:** 4
**Confidence:** 3

**Summary:**

The paper studies zero-shot Prediction-Powered Causal Inference (PPCI), where a pretrained model predicts outcomes on a new experiment without any labeled data. It formalizes validity and introduces a novel “Causal Lifting” constraint, $I(\phi(X), Z \mid Y) = 0$ to ensure that representations carry no spurious signals. A practical instantiation - Deconfounded Empirical Risk Minimization reweights training samples to enforce lifting. Experiments on a replicated ISTAnt ecological dataset demonstrate that DERM is the only method to yield valid zero-shot ATE estimates across diverse pretrained backbones and shifts.

**Questions:**

Can the authors comment on my concerns?

**Ethical Concerns:**

["NO or VERY MINOR ethics concerns only"]

**Final Justification:**

After the rebuttal as well as looking at other reviewers comments, I decided to recommend borderline accept.

**Limitations:**

Yes.

**Quality:**

3

**Strengths And Weaknesses:**

The paper tackles a timely, scientifically motivated problem via its formalization of PPCI and causal lifting constraints, while DERM offers a straightforward, model-agnostic recipe that achieves zero-shot ATE validity where baselines including ERM, IRM, etc fail.

Weakness:

- It is very difficult to assess whether the implementation/tuning of different methods are correctly aligned. In Fig 2, all failure cases for ERM, IRM and vRex are all from ViT-S and -L models. Is it a coincidence? I think it might be possible that there are confounding among the performance of different methods. For example, I noticed that the Adam optimizer uses beta1=0.9, beta2=0.9, which arguably is different from common optimal settings and can bias the performance of certain model-method pairs. Moreover, there are certain parameters that are not properly ablated (eg, training epochs, batchsize etc) . In my practical experience those parameters can result in big swings in final results (ie relative rankings) for causal inference algorithms and might induce significant confounding effect for evaluation.

- The other minor thing is that the work relies heavily on the assumption that the annotation mechanism truly transfers between experiments - which is an unverifiable condition. Furthermore, not everyone agree that the prediction/annotation problem is an anti-causal problem. In fact some often think of X (recordings) and Y (human labels) as two different representation spaces projected from nature variables via causal mechanism and cognitive/measurement mechanism). In this causal model, it is a strong assumption that the full mechanism between X and Y can be transferred across environments.

But in general I like this paper.

---

> ### Author Rebuttal · Authors · 2025-07-30
>
> We thank the reviewer @7mz6 for the careful and hands-on feedback. Overall, the general message to “_tackle a timely, scientifically motivated problem via its formalization_” and “_offering a straightforward, model-agnostic_” solution seems delivered. Two concerns are raised, which we respond to here and plan to adapt the final version of the manuscript accordingly.
>
> **Model selection** (Weakness 1)
>
> - We considered the best-performing model hyperparameters (which encoder tokens to use, head architecture, batch-size, etc.) on ISTAnt according to the extensive ablation study in Cadei et al. (2024). These were the only arbitrary choices and were performed using ERM, so they may **at most favour the ERM** baseline, which still drastically fails. For each method (ERM, vREX, IRM, ours), we tuned the learning rate and invariance regularizer of IRM and vREX for enough epochs (15) to allow convergence. We performed a grid search, discarding hyperparameters that led to factual outcome prediction accuracy smaller than 0.8 on a validation set from the training experiment (1000 samples). Next, we selected for each method the hyperparameters leading to the smallest Treatment Effect Bias [Cadei et al., 2024] estimated on the full prediction-powered training experiment (i.e., no leaking from the target distribution). We plan to clarify these steps in the final version of the paper, in Appendix B.2.
> - Adam's second moment coefficient is a **typo**, and we confirm we used instead default values ($\beta_1=0.9$, $\beta_2=0.999$), as shared in the code.
> - It is true that some encoders are more performant/confounded than others, also in agreement with the original paper from Cadei et al. (2024), yet our approach always succeeded in enforcing valid estimates. New benchmarks are required for a more exhaustive comparison among pre-trained models and inner causal entanglement evaluation, i.e., $I(\phi(X), Z|Y)$, but this is out of the scope of this paper.
>
> **Annotation Mechanism Assumption** (Weakness 2)
> - Annotation Mechanism invariance is **assumed anyway by domain (human) experts**, even without relying on artificial predictions. Taking our application as an example, experimental ecologists practically learn by experience how to recognize behaviours, and then assume it is transferable when they run new experiments and collect the same labels. Let’s further observe that in practice, we are never relying on their causal relationship, i.e., between the high-dimensional recordings (X) and the outcome behaviours (Y).
>
>
> [1] Cadei, Riccardo et al. “Smoke and Mirrors in Causal Downstream Tasks” NeurIPS 2024

---

> > ### Comment · Reviewer_7mz6 · 2025-08-04
> >
> > Thank you for your response. I will keep my score / or update in a positive direction.

---

> > > ### Author Response · Authors · 2025-08-05
> > >
> > > **Follow-up Method Robustness** (*Weakness 1*)
> > >
> > > Based on the discussion with reviewer *`@xo43`*, we extended the method robustness ablation, presenting new results on CausalMNIST comparing different downstream causal estimators (see Table 1-2 reported here). As expected, every tested causal estimator is invalidated by biased prediction-powered samples, regardless of its robustness to confounder effects. The mitigation we presented for Prediction Powered Causal Inference (causal lifting) relies instead on the predictive model's conditional calibration, whose benefits are not specific to AIPW.
> > >
> > > **Results:**
> > >
> > > - *Table 1*: Average Treatment Effect Bias (± standard deviation over 50 experiment repetitions) over prediction-powered samples by a predictive model trained by **ERM**. As for AIPW, other causal estimators fail in estimating the effect, i.e., TEB$\gg 0$, on biased prediction-powered samples.
> > >
> > >
> > > | Estimator       | In-Distr.       | Soft-Shift (linear effect) | Hard-Shift (linear effect) | Soft-Shift (nonlinear effect) | Hard-Shift (nonlinear effect) |
> > > |-----------------|------------------|-----------------------------|-----------------------------|-------------------------------|-------------------------------|
> > > | AIPW            | 0.00 ± 0.02      | 0.86 ± 0.14                | 1.05 ± 0.15                | 0.64 ± 0.18                   | 0.85 ± 0.17                   |
> > > | X-Learner       | 0.01 ± 0.02      | 0.82 ± 0.16                | 0.92 ± 0.15                | 0.68 ± 0.19                   | 0.79 ± 0.18                   |
> > > | BART            | 0.01 ± 0.02      | 0.81 ± 0.19                | 0.89 ± 0.13                | 0.59 ± 0.21                   | 0.75 ± 0.16                   |
> > > | Causal Forest   | 0.01 ± 0.02      | 0.82 ± 0.13                | 0.90 ± 0.11                | 0.64 ± 0.16                   | 0.76 ± 0.12                   |
> > >
> > >
> > >
> > > - *Table 2*: Average Treatment Effect Bias (± standard deviation over 50 experiment repetitions) over prediction-powered samples by a predictive model trained by **DERM (ours)**. Our approach generalizes to other downstream causal estimators, i.e., TEB$\approx 0$ (under soft shifts).
> > >
> > > | Estimator       | In-Distr.       | Soft-Shift (linear effect) | Hard-Shift (linear effect) | Soft-Shift (nonlinear effect) | Hard-Shift (nonlinear effect) |
> > > |-----------------|------------------|-----------------------------|-----------------------------|-------------------------------|-------------------------------|
> > > | AIPW            | 0.10 ± 0.07      | 0.14 ± 0.14                | 0.75 ± 0.05                | 0.08 ± 0.27                   | 0.45 ± 0.12                   |
> > > | X-Learner       | 0.32 ± 0.27      | 0.00 ± 0.25                | 0.69 ± 0.12                | 0.05 ± 0.27                   | 0.53 ± 0.14                   |
> > > | BART            | 0.19 ± 0.14      | 0.07 ± 0.14                | 0.68 ± 0.15                | 0.06 ± 0.18                   | 0.52 ± 0.18                   |
> > > | Causal Forest   | 0.22 ± 0.19      | 0.11 ± 0.13                | 0.67 ± 0.09                | 0.03 ± 0.24                   | 0.50 ± 0.15                   |

---

### Official Review · Reviewer_xo43 · 2025-07-03

**Clarity:** 3
**Significance:** 3
**Originality:** 3
**Rating:** 5
**Confidence:** 3

**Summary:**

This paper proposes a new framework for Prediction-Powered Causal Inference (PPCI), which enables zero-shot estimation of treatment effects from high-dimensional, unlabeled data using pre-trained neural predictors.

The core idea is to reuse models trained on similar experiments to infer causal quantities in new experiments without additional labeling.

The authors identify conditional calibration as a key statistical property that such predictors must satisfy. However, standard empirical risk minimization (ERM) often fails to produce conditionally calibrated models due to spurious correlations in the data. To address this, the paper introduces the concept of Causal Lifting, a constraint that prevents outcome predictors from leveraging spurious cues related to treatment or experimental settings. They implement this constraint via a practical training method called DERM, which reweights training samples to eliminate correlations between experimental settings and outcomes.

**Questions:**

The paper does not compare with well-established causal effect estimation methods such as TARNet, CEVAE, BART, or X-learner, which may perform competitively under different assumptions. Could the authors justify this omission and clarify whether such methods are applicable or complementary in the context of zero-shot PPCI?

**Ethical Concerns:**

["NO or VERY MINOR ethics concerns only"]

**Final Justification:**

Authors added more experiments to address my concerns.

**Limitations:**

Yes

**Quality:**

3

**Strengths And Weaknesses:**

Strengths:

- The paper establishes rigorous conditions for when predictors yield valid causal estimates, notably through the conditional calibration property and the causal lifting constraint. These theoretical insights connect predictive modeling to causal identifiability.

- The authors propose DERM as a simple and effective method to enforce the causal lifting constraint. It is well-motivated and practically implementable through reweighting based on outcome variance.

- The paper reports the first valid zero-shot PPCI result on ISTAnt, showcasing a significant advancement over prior methods like ERM, IRM, and v-REx.

- In Table 1, although DERM exhibits slightly higher error in the in-distribution setting compared to ERM and other baselines, it significantly outperforms them in OoD environments. This trade-off is intentional and aligned with the paper’s goal of enabling zero-shot causal inference across unseen experimental settings. The results validate that DERM effectively mitigates reliance on spurious correlations present in the source domain, which are often exploited by baselines to achieve lower in-distribution error but lead to invalid causal conclusions under distribution shifts.

Weaknesses:

1/ The framework relies on key assumptions: conditional calibration and the causal lifting constraint. These assumptions cannot be verified on the target dataset. This limits practical confidence in the causal validity of the estimates in real-world deployments.

2/ From my understanding, the effectiveness of DERM depends on Eq. (9) (full support of outcomes across covariates). If this condition fails (as explored in the synthetic "hard shift" setting), DERM may still pick up spurious correlations, potentially harming causal estimation.

3/ While comparisons are made with ERM, IRM, and v-REx, the paper does not evaluate against classical or modern causal effect estimation methods like TARNet, CEVAE, BART, or X-learner, which could offer strong baselines under different assumptions.

4/ Typos (does not effect my evaluation):

Line 309: convolutiona

Line 306: experrimental

Table 1 caption: (soft, hard shifts). we report

---

> ### Author Rebuttal · Authors · 2025-07-30
>
> We thank the reviewer @xo43 for the detailed feedback. Overall, the general message to “_connect predictive modeling to causal identifiability_”, detecting “_rigorous conditions_” and proposing a “_simple and effective method_”  seems delivered. The concerns relate to (i) assumptions test and justification, and (ii) causal model baselines. We comment on the former, while we disagree on the latter for factual points, and plan to adapt the final version of the manuscript with corresponding clarifications.
>
> (i) **Assumptions clarifications** (Weaknesses 1 and 2)
>
> - _Conditional Calibration_
>   - Conditional Calibration on the target distribution is not an assumption, but a sufficient condition for valid causal inference. It can be enforced and tested on the training distribution, and we show in Thm 3.1 under which assumptions it generalizes to a similar experiment even if out-of-distribution.
>
> - _Causal Lifting Constraint_
>   - The key assumption enabling Thm. 3.1 is that the Causal Lifting constraint holds on the target distribution. We can train and test whether this constraint holds on the training, but the reviewer is correct that, in general, it may not transfer.
>   - Between the two evils, we propose to enforce the Causal Lifting constraint and not directly Conditional Calibration. Indeed, a model optimized for standalone Conditional Calibration may still fail to transfer when dealing with new effect modifiers with similar appearance but different effects from the ones in the training experiment (see Example 2). This approach is similar in spirit to the considered Domain Generalization baselines, i.e., vREx and IRM, enforcing invariant prediction performances (ideally unbiased) varying the experiment settings. Instead, the Causal Lifting constraint robustly prevents such shortcuts directly in the representation space by ignoring any source of possible spurious correlations.
>   - Ultimately, it is true that the Causal Lifting constraint is not a testable assumption on the target distribution (at least zero-shot). However, we believe that the approach has merit as we demonstrated empirically just enforcing the Causal Lifting constraint on the training experiment and validly estimating the treatment effect on a similar target experiment with different treatments and recording platforms.
>
> - _Full Support Assumption_
>   - DERM is our practical implementation of the Causal Lifting constraint, by sample reweighting, relying on the full support of outcomes across covariates, i.e., Eq. 9. This is easy to test, e.g., check that the behaviour of interest is happening at least once for each experimental setting; still, we agree that there could be scenarios where such a support assumption is challenged/violated, so losing its guarantees (we acknowledge it in the Conclusion). In practice, DERM still outperforms ERM, IRM,vREx, even in such a regime (see Hard Shifts experiments on CausalMNIST), and the assumption is not violated in our real-world application.
>
>
>
> (ii)  **Comparison with other Causal Estimators** (Weakness 3)
>
> - The reviewer wonders if additionally benchmarking our pipeline for PPCI, i.e., causal lifting of foundational model and AIPW estimator, against standalone established causal effect estimators. However, it is not possible by design, since all the proposed causal estimators rely on the factual outcomes of the target experiment, which are not observed directly in PPCI, and only observed as raw measurements, e.g., pixels. More precisely:
>   - In XLearner, E[Y|T=t, W=w] cannot be estimated on the target experiment because Y is not observed. On the training population, the corresponding conditional expectation can be completely different, as the effect can vary.
>   - TARNet requires unavailable (target) factual outcome supervision for the network training. Possible variants, additionally relying on a pre-trained encoder, may be considered for fine-tuning on the training distribution (e.g., concatenating X and W) but without generalization guarantees (similar in spirit to our ERM baseline).
>   - Similarly, BART requires unavailable (target) factual outcome supervision for each regression. Even if trained on the training distribution, it has no guarantee to generalize to a target experiment with different causal effects, and anyway, it wouldn’t scale on complex data structures, e.g., images.
>   - CEVAE still requires unavailable (target) factual outcome supervision for training. Even if trained on the training distribution, it has no guarantee to generalize, and anyway, it is an overkill to assume unobserved confounders when they are actually observed (W) and potentially controlled too, e.g., RCT.

---

> > ### Comment · Reviewer_xo43 · 2025-08-04
> >
> > Thank you for the detailed and thoughtful response. I appreciate the clarifications provided on both the assumptions and the causal baselines.
> >
> > Regarding my earlier comment on comparisons to traditional causal estimators, I realize I may not have been entirely clear. My intention was not to suggest that these methods should be applied directly to the unobserved outcomes in the target domain, but rather to ask whether they could be applied to the model-predicted outcomes  g(X) as a way to demonstrate their failure in this setting. I believe such a comparison (while not causally valid ) can serve as an illustrative baseline to highlight the necessity of principled constraints like causal lifting.
> >
> > Nevertheless, I maintain my positive opinion about the paper.

---

> > > ### Author Response · Authors · 2025-08-05
> > >
> > > **Follow-up Comparison with other Casual Inference estimators** [*new ablation*]:
> > >
> > >
> > > We thank the reviewer for the clarification and constructive engagement. Following the suggestion, we additionally repeated the CausalMNIST experiments (to guarantee valid uncertainty quantification by resampling), replacing the AIPW estimator with the proposed estimators (additionally including Causal Forest). We ignored neural-network-based estimators, i.e., TARNet and CEVAE (they are overkill for the task complexity, i.e., RCT with 2 observed covariates).
> > >
> > > **Take-away**: Every tested causal estimator is invalidated by biased prediction-powered samples, regardless of its robustness to confounder effects. The mitigation we presented for Prediction Powered Causal Inference (causal lifting) relies instead on the predictive model's conditional calibration, whose benefits are not specific to AIPW.
> > >
> > > **Results**:
> > >
> > > - *Table 1*: Average Treatment Effect Bias ($\pm$ standard deviation over 50 experiment repetitions) over prediction-powered samples by a predictive model trained by ERM. As for AIPW, other causal estimators fail in estimating the effect, i.e., TEB$\gg0$, on biased prediction-powered samples.
> > >
> > >
> > > | Estimator       | In-Distr.       | Soft-Shift (linear effect) | Hard-Shift (linear effect) | Soft-Shift (nonlinear effect) | Hard-Shift (nonlinear effect) |
> > > |-----------------|------------------|-----------------------------|-----------------------------|-------------------------------|-------------------------------|
> > > | AIPW            | 0.00 ± 0.02      | 0.86 ± 0.14                | 1.05 ± 0.15                | 0.64 ± 0.18                   | 0.85 ± 0.17                   |
> > > | X-Learner       | 0.01 ± 0.02      | 0.82 ± 0.16                | 0.92 ± 0.15                | 0.68 ± 0.19                   | 0.79 ± 0.18                   |
> > > | BART            | 0.01 ± 0.02      | 0.81 ± 0.19                | 0.89 ± 0.13                | 0.59 ± 0.21                   | 0.75 ± 0.16                   |
> > > | Causal Forest   | 0.01 ± 0.02      | 0.82 ± 0.13                | 0.90 ± 0.11                | 0.64 ± 0.16                   | 0.76 ± 0.12                   |
> > >
> > >
> > >
> > > - *Table 2*: Average Treatment Effect Bias ($\pm$ standard deviation over 50 experiment repetitions) over prediction-powered samples by a predictive model trained by DERM (ours). Our approach generalizes to other downstream causal estimators, i.e., TEB$\approx 0$ (under soft shifts).
> > >
> > > | Estimator       | In-Distr.       | Soft-Shift (linear effect) | Hard-Shift (linear effect) | Soft-Shift (nonlinear effect) | Hard-Shift (nonlinear effect) |
> > > |-----------------|------------------|-----------------------------|-----------------------------|-------------------------------|-------------------------------|
> > > | AIPW            | 0.10 ± 0.07      | 0.14 ± 0.14                | 0.75 ± 0.05                | 0.08 ± 0.27                   | 0.45 ± 0.12                   |
> > > | X-Learner       | 0.32 ± 0.27      | 0.00 ± 0.25                | 0.69 ± 0.12                | 0.05 ± 0.27                   | 0.53 ± 0.14                   |
> > > | BART            | 0.19 ± 0.14      | 0.07 ± 0.14                | 0.68 ± 0.15                | 0.06 ± 0.18                   | 0.52 ± 0.18                   |
> > > | Causal Forest   | 0.22 ± 0.19      | 0.11 ± 0.13                | 0.67 ± 0.09                | 0.03 ± 0.24                   | 0.50 ± 0.15                   |
> > >
> > >
> > >
> > > Happy to include this ablation in the Appendix of the final version, hoping to clarify the dependence among the two steps (factual outcome model training and causal effect estimation).

---

> > > > ### Comment · Reviewer_xo43 · 2025-08-06
> > > >
> > > > Thank you for the experiments. I have updated the score accordingly.

---

### Decision · Program_Chairs · 2025-09-17

**Decision:**

Accept (poster)

**Comment:**

This paper introduces a new framework for Prediction-Powered Causal Inference (PPCI), which uses pre-trained neural networks to estimate treatment effects from high-dimensional, unlabeled data. The core concept is to reuse models from similar experiments to infer causal relationships in new, unlabeled datasets. The authors identify conditional calibration as a crucial statistical property for these predictors. However, standard training methods often fail to achieve this due to spurious correlations in the data. To solve this, the paper proposes Causal Lifting, a constraint that prevents outcome predictors from relying on these spurious cues. They implement this constraint using DERM, a practical training method that reweights training samples to eliminate correlations between experimental settings and outcomes.

The reviews are overall positive, with all reviewers leaning toward acceptance after the author-reviewer discussion phase. However, some reviewers still found the paper difficult to read and understand, suggesting a lack of clarity.

Given its novelty and technical contributions, I recommend its acceptance for publication at NeurIPS as a poster. I encourage the authors to further improve the clarity of presentation in the camera-ready version.